# Non-canonical Metatranscriptomic analysis of COVID-19 and Dengue reveals an expanded microbial and AMR landscape in COVID-19 mortality patients

Aanchal Yadav[1,2☯], Raiyan Ali[1☯], Priti Devi[1,2], Pallawi Kumari[1,3], Jyoti Soni[1,2], Garima [1,2], Bansidhar Tarai[4], Sandeep Budhiraja[4], Uzma Shamim[1,5]*, Rajesh Pandey [1,2‡]*

**1** Division of Immunology and Infectious Disease Biology, INtegrative GENomics of HOst-PathogEn (INGEN-HOPE) Laboratory, CSIR-Institute of Genomics and Integrative Biology (CSIR-IGIB), Delhi, India, **2** Academy of Scientific and Innovative Research (AcSIR), Ghaziabad, Uttar Pradesh, India, **3** Indraprastha Institute of Information Technology (IIIT), New Delhi, India, **4** Max Super Speciality Hospital (A Unit of Devki Devi Foundation), Max Healthcare, Delhi, India, **5** Ashoka University, Sonipat, Haryana, India

☯ These authors contributed equally to this work.
‡ Lead contact.
* rajeshp@igib.in, rajeshp.igib@csir.res.in (RP); uzma.shamim@igib.in (US)

## Abstract

AMR is a growing concern in viral infections, where microbiome shifts contribute to resistance gene dissemination. While dengue and COVID-19, caused by ssRNA viruses, are not bacterial-driven, their resistome and microbial communities influence disease progression and AMR burden. This study analyzes the resistome and microbiome in 251 COVID-19 and 112 dengue patients using non-canonical metatranscriptomics. By mapping antimicrobial resistance genes (ARGs) and their transcriptionally active microbes (TAMs) hosts, we uncover greater ARG burden in COVID-19, particularly during mortality, with a diverse set of associated TAMs compared to dengue. MDR genes were prevalent, with beta-lactamase ARGs commonly detected in both infections. COVID-19 exhibited higher carbapenemase resistance genes (*NDM, OXA, VIM*), while dengue was associated with TEM variants. *Escherichia coli* and *Klebsiella pneumoniae* were dominant ARG hosts, with *Acinetobacter baumannii* in COVID-19 mortality and *Bacillus cereus* in severe dengue. These findings highlight resistome dynamics and emphasize the need for AMR surveillance in high-burden infections.

which permits unrestricted use, distribution, and reproduction in any medium, provided the original author and source are credited.

**Data availability statement:** Sequence data used in this study are publicly available in NCBI SRA under the accession number with the BioProject ID: PRJNA676016, PRJNA678831, PRJNA868733, PRJNA952815 (COVID-19), and PRJNA1071729, PRJNA1279769 (Dengue).

**Funding:** This study received financial support from Bill and Melinda Gates Foundation (BMGF), (Grant number - INV-033578) and Rockefeller Foundation, (grant number—2021 HTH 018) awarded to RP. The funders had no role in study design, data collection and analysis, decision to publish, or preparation of the manuscript.

**Competing interests:** The authors have declared that no competing interests exist.

## Author summary

Antibiotics are frequently administered to manage secondary complications in viral infections like COVID-19 and dengue, yet their broad usage can unintentionally amplify the resistome—the pool of antimicrobial resistance genes (ARGs)—within transcriptionally active microbes (TAMs). This aspect is often underexplored in non-bacterial infections. In this study, we analyzed human-unmapped reads from the transcriptomic data of 251 COVID-19 and 112 dengue patients to explore the active microbial community and their associated resistome profiles. Our findings reveal that both diseases harbor distinct and overlapping ARGs across multiple drug classes, including multidrug resistance (MDR) genes such as those conferring resistance to β-lactam antibiotics, which were detected in 49.5% of COVID-19 and 56.5% of dengue patients. Key resistant microbes included *Escherichia coli*, *Klebsiella pneumoniae*, and *Acinetobacter baumannii* in COVID-19, and *Bacillus cereus* in severe dengue. Notably, COVID-19 samples exhibited greater microbial diversity and ARG abundance compared to dengue, suggesting a stronger resistome influence on disease severity in COVID-19. These findings highlight the need to integrate resistome surveillance into the clinical management of viral infections, guiding more informed and targeted antibiotic use.

## Introduction

Infectious diseases remain a major global health challenge, with bacteria, fungi, protozoa, helminths, and viruses continuing to drive morbidity and mortality. Among viral infections, outbreaks of COVID-19 and dengue have repeatedly strained healthcare systems, particularly in the tropical and subtropical regions [1]. Both diseases, caused by single-stranded RNA viruses—SARS-CoV-2 and the dengue virus, respectively—differ in transmission routes and clinical manifestations but share a critical consequence: high rates of secondary bacterial infections and extensive antibiotic use [2,3].

The overuse of antibiotics, both in clinical settings and beyond, accelerates antimicrobial resistance (AMR), a crisis projected to cause 10 million deaths annually by 2050 [4]. While bacterial resistance to antibiotics is well-documented, the role of viral infections in shaping the antimicrobial resistance landscape remains underexplored. Emerging evidence suggests that viral infections can influence the resistome—the collection of antimicrobial resistance genes (ARGs) within a microbial community—by altering host immune responses, disrupting microbiome composition, and driving excessive antibiotic use [5]. For instance, widespread antibiotic use/misuse in the COVID-19 patients has contributed to AMR, while in dengue, early misdiagnosis often leads to unnecessary antibiotic prescriptions, further disrupting microbial balance and fostering resistance [6,7].

Understanding the microbiome resistome—the interplay between commensal and pathogenic bacteria in harboring and transferring ARGs—is critical from a One Health perspective. Elucidating the abundance and composition of these microbial communities, specially the metabolically or transcriptionally active microbes (TAMs), is

crucial for altering treatment strategies to mitigate disease severity and dissemination of resistance. Previous work by our group described how these active microbial species in patients with severe COVID-19 and dengue were often linked to pathogenic TAMs [8–12]. However, the contribution of these TAMs to the functional resistome remains largely unexplored. Given that TAMs play a critical role in shaping the host-pathogen interface, their involvement in the dissemination of AMR genes warrants further investigation. Understanding how TAMs influence the active resistome during these infections is crucial for elucidating the mechanisms of AMR gene transmission, persistence, and expression in the host.

Conventional methods for studying AMR, such as culture-based antimicrobial susceptibility testing, have limitations in capturing the full complexity of microbial communities, particularly non-culturable or viable-but-nonculturable (VBNC) cells [13]. Advances in genome sequencing and meta-transcriptomics now provide powerful tools for de-novo identifications of ARGs, their host bacteria, and functional resistance pathways with greater sensitivity and accuracy [14,15]. In this study, we employed a previously lab used "non-canonical meta-transcriptomics" approach, which we define as the repurposing of host total RNA-seq data—originally generated for the host transcriptomics—to identify and quantify TAMs. Unlike conventional metagenomics or metatranscriptomics, which involve microbial DNA/RNA enrichment, our method starts from native clinical samples based on disease manifestation site (e.g., blood or nasopharyngeal swabs for Dengue and COVID-19) and removes host-aligned reads computationally to analyze microbial expression. This strategy enables unbiased profiling of active microbial communities without introducing potential experimental biases from the enrichment protocols, which can distort transcript abundance in low-biomass, host-dominated samples.

Building on our previous research on TAMs that linked severe dengue to greater microbial diversity, enrichment of opportunistic species like *Bacillus cereus* and *Burkholderia pseudomallei*, and elevated microbial energy metabolism, we also reported variant-specific shifts in TAMs during SARS-CoV-2 infection—highlighting *Acinetobacter baumannii* and *Pseudomonas aeuroginosa* as a potential biomarker associated with immune response and clinical severity in Delta VOC [10,12]. We now expand this investigation, employing a non-canonical meta-transcriptomics approach to identify and compare the resistome and microbiome profiles of COVID-19 and dengue patients, providing a new perspective of AMR evolution in infectious disease settings. By mapping ARGs and their transcriptionally active host bacteria, we identify shared and disease-specific ARGs for early detection of emerging resistance patterns across the clinical environments. Our analysis reveals an overall higher ARG burden in the COVID-19 patients, particularly in the mortality patients, and a dominance of multidrug resistance genes in both the infections—highlighting the consequences of antibiotic misuse. These insights from the clinical hospital admitted patients for both the diseases, along with the detailed clinical data vis-à-vis their disease severity sub-phenotypes have direct implications for clinical management strategies aimed at mitigating AMR in the high-burden infections.

## Results

### Study design and sequencing statistics

The present study was designed to investigate the RNA virus infected host's microbiome, especially TAMs, for the potential reservoir of ARGs it carries and its comparative and distinct patterns in relation to the viral diseases, COVID-19 and dengue (Fig 1a). We included 363 patients, COVID-19 ($n=251$) and dengue ($n=112$) in this study, wherein COVID-19 patients were confirmed via RT-PCR, and dengue cases tested positive for the NS1 antigen. Based on clinical severity, COVID-19 patients fell into following groups: mild ($n=128$), moderate ($n=33$), severe ($n=24$) and mortality ($n=60$). Similarly, the dengue had subgroups of mild ($n=45$), moderate ($n=46$), and severe ($n=21$). No mortality cases were reported in our dengue study samples.

Holo transcriptome sequencing on the Illumina NextSeq 2000 platform was done. An average of 18,411,821 and 18,784,780 reads were generated for the COVID-19 and dengue, respectively. Post-quality control, reads were classified as human-mapped (10,126,501 in COVID-19, 17,150,504 in dengue) and unmapped (8,285,319 in COVID-19, 1,634,275 in dengue). As evident, the unmapped reads captured for COVID-19 were considerably higher in the COVID-19 than

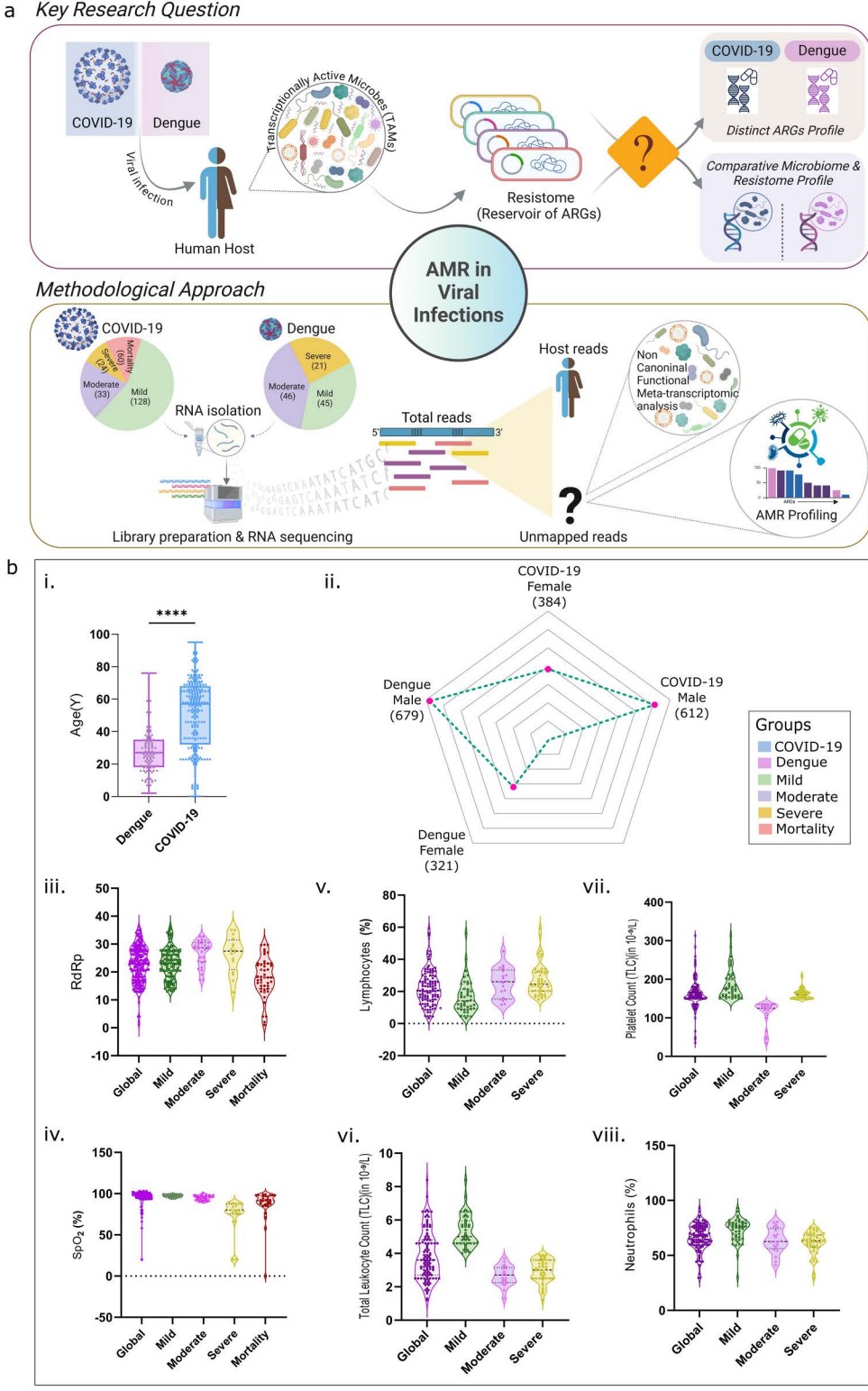

**Fig 1. Overview of study design and clinical data. (a)** Schematic representation of the principal research problem and methodology, illustrating the distinction between the COVID-19 and dengue microbiome and resistome using a holo-transcriptomic sequencing approach. *Created in BioRender. Devi, P. (2025).* **(b)** Clinical data visualization, including: **(i)** Bar plot depicting age distribution of COVID-19 and dengue patients, **(ii)** Spider plot

illustrating gender-based patient distribution, **(iii)** *RdRp* levels, and **(iv)** *SpO₂* levels across the different COVID-19 severity groups; **(v)** Lymphocyte count, **(vi)** total leukocyte count (TLC), **(vii)** platelet count, and **(viii)** neutrophil levels, highlighting significant CBC parameters across the dengue severity groups.

dengue, owing to the site of viral infection, nasopharyngeal for COVID-19 and blood for dengue. We observed clear disease-specific differences in the microbial signal, with ~58% host reads in the COVID-19 samples and ~92% in the dengue samples, supporting the biological relevance of the leftover microbial reads analyzed in this study. Yet, the proportion of human (H) to non-human (nH) reads remained consistent across severity groups, i) COVID-19: mild (63.7% H, 36.3% nH), moderate (63.9% H, 36.1% nH), and severe (70.1% H, 29.9% nH), ii) dengue: mild (91.7% H, 8.3% nH), moderate (91.3% H, 8.7% nH), and severe (90.3% H, 9.7% nH). Interestingly, mortality group patients exhibited contrast profiles with 37.7% H and 62.3% nH reads. Subsequently, the non-human reads were analyzed for transcriptionally active microbial content, by virtue of they being captured through RNA-seq, and downstream functional analysis. The methodological approach is illustrated in Fig 1a.

## Clinical parameters and statistical insights

In our study, all samples from both the COVID-19 and dengue patient groups were collected on the first day of hospital reporting, prior to admission and before the initiation of any in-hospital treatment. However, pre-hospital antibiotic usage was not prospectively recorded. Partial data on antibiotic use were available only for the COVID-19 patients (n = 145 across severity groups: 88- mild, 28- moderate, 14- severe, and 15- mortality patients) and are provided in S1 Table. Further, clinical parameters were evaluated to assess disease severity (Table 1). A bar plot (Fig 1b.i) illustrates the distribution of cases, showing a higher prevalence of COVID-19. A spider plot (Fig 1b.ii) revealed male predominance in both the groups—61.2% in COVID-19 and 67.9% in dengue. For COVID-19, $SpO_2$ (peripheral oxygen saturation) and *RdRp gene* (RNA-dependent RNA polymerase) were the key severity indicators. The global median of *RdRp* was 23.06, with mild (23.12), moderate (28.57), severe (27.46), and mortality (18.00) groups showing significant differences (Fig 1b.iii). $SpO_2$ levels were lowest in severe (80) and mortality (91) patients compared to the mild (97) and moderate (95), with significant statistical differences (Fig 1b.iv). In contrast, other severity-modulating factors, including age, gender, and comorbidities, were analyzed across both infections and their respective severity classes. Gender showed no significant differences across severity groups in either the COVID-19 or dengue cohorts. Age did not demonstrate significant differences across severity classes in the dengue cohort. However, in the COVID-19 cohort, age was significantly associated with severity when comparing all severity classes collectively. Notably, individual group comparisons, such as the moderate group (median age: 62) and the mortality group (median age: 65), revealed similar age distributions despite differing severity levels. This suggests that while age is associated with disease severity, it does not solely account for the observed differences in severity, indicating the influence of additional factors beyond age alone. Similarly, comorbidities among COVID-19 patients were not significantly associated with disease severity (Table 1). Comorbidity data for the dengue patients, however, were not available for analysis. In dengue, a complete blood count (CBC) analysis of 10 parameters revealed significant differences in the *TLC,* platelet count, neutrophils, and lymphocytes. Mild patients had elevated TLC, platelet count, and neutrophils, while moderate and severe showed higher lymphocyte levels (Fig 1b.v, vi, vii, viii). In COVID-19, factors like $SpO_2$ and RdRp play a critical role in determining disease severity and outcomes, while in dengue, differences in CBC parameters across the severity groups indicate the involvement of different factors, with the microbial aspect being one potential area of influence under investigation. The observed differences in the CBC parameters provide a basis for investigating how TAMs interact with immune and physiological changes across the dengue severity subgroups.

**Table 1. COVID-19 and Dengue patient's clinical parameters.**

| COVID-19 | COVID-19 Clinical Parameters | Mild (n=97) | Moderate (n=95) | Severe (n=80) | Mortality (n=91) | p-values | |
|---|---|---|---|---|---|---|---|
| | Age | 46 | 62 | 58 | 65 | <0.0001 | b |
| | SpO2 | 97 | 95 | 80 | 91.5 | <0.0001 | b |
| | RdRp | 23.12 | 28.57 | 27.46 | 18 | <0.0001 | b |
| | Hospital stay (days) | 6 | 12 | 10 | 9.5 | 0.0061 | a |
| | Gender F\|M | 54\|74 | 11\|22 | 9\|15 | 20\|39 | 0.64934 | a |
| Comorbidities | Asthma | 1 (0.78) | 1 (3.03) | 0 | 0 | .298309 | a |
| | Diabetes | 15 (11.72) | 5 (15.15) | 6 (24) | 3 (5) | .907046 | a |
| | Heart disease | 3 (2.34) | 2 (6.06) | 1 (4) | 0 | .540653 | a |
| | Hypertension | 21 (16.41) | 8 (24.24) | 2 (8) | 4 (6.67) | .082804 | a |
| | Thyroid | 2 (1.56) | 0.00 | 2 (8) | 1 (1.67) | .12262 | a |
| | kidney | 5 (3.91) | 4 (12.12) | 0 | 0 | .067007 | a |
| | lungs | 2 (1.56) | 2 (6.06) | 0 | 0 | .138816 | a |
| | Others | 7 (5.47) | 4 (12.12) | 2 (8) | 2 (3.33) | .365068 | |
| DENGUE | Dengue Clinical Parameters | Mild (n=45) | Moderate (n=46) | Severe (n=21) | | p-values | |
| | Age | 27 | 26 | 35 | | 0.13963 | b |
| | Gender F\|M | 15\|30 | 17\|29 | 4\|17 | | 0.338071 | a |
| | NS1 antigen | 3.5 | 3.5 | 3.5 | | 0.7536 | b |
| | CBC | | | | | | |
| | Total Leucocyte Count (TLC) | 5 | 3 | 2.7 | | <.001 | b |
| | RBC Count | 4.8 | 4.835 | 4.78 | | 0.7556 | b |
| | Packed Cell, Volume | 41.2 | 42.25 | 43.9 | | 0.184 | b |
| | Platelet Count | 175 | 160 | 125 | | <.001 | b |
| | Neutrophils | 75.9 | 63.3 | 62.6 | | <.001 | b |
| | Lymphocytes | 14.1 | 24.5 | 26 | | <.001 | b |
| | Monocytes | 9.8 | 10 | 9 | | 0.31923 | b |
| | Eosinophils | 0.2 | 0.2 | 0.5 | | 0.32048 | b |
| | Basophils | 0.5 | 0.6 | 0.5 | | 0.6906 | b |
| | Hb | 13.6 | 13.9 | 14.1 | | 0.28362 | b |
| | LFT | | | | | | |
| | Total Protein | 7.3 | 6.7 | 6.7 | | 0.02715 | b |
| | Albumin | 4.4 | 4 | 4 | | 0.0449 | b |
| | Globulin | 3.1 | 3 | 2.65 | | 0.0557 | b |
| | A.G. ratio | 1.4 | 1.3 | 1.55 | | 0.18929 | b |
| | Bilirubin (Total) | 0.7 | 0.4 | 0.75 | | 0.0184 | b |
| | Bilirubin (Direct) | 0.135 | 0.09 | 0.175 | | 0.01765 | b |
| | Bilirubin (Indirect) | 0.56 | 0.31 | 0.555 | | 0.0277 | b |
| | SGOT- Aspartate Transaminase (AST) | 41 | 59 | 97.5 | | 0.0266 | b |
| | SGPT- Alanine Transaminase (ALT) | 42 | 41 | 66 | | 0.1314 | b |
| | AST/ALT Ratio | 0.93 | 1.48 | 1.705 | | 0.0767 | b |
| | Alkaline Phosphatase | 67 | 80 | 82.5 | | 0.609 | b |
| | GGTP (Gamma GT), Serum | 37 | 19 | 49 | | 0.0786 | b |

a-chi square, b-Kruskal Wallis.

## Distribution and abundance of antimicrobial resistance genes (ARGs) in COVID-19 and Dengue patients

**Resistome analysis reveals higher ARGs load in COVID-19 and shared MDR dominance across infections.** In this study, we explored the resistome—the collection of antimicrobial resistance genes (ARGs)— in patients with COVID-19 and dengue, uncovering intriguing patterns in how microbes and their resistome would impact the antibiotics usage given as part of the clinical management plan for preventing secondary infections. Our comprehensive holo-transcriptomic analysis of 251 COVID-19 and 112 dengue patients during the early stages of infection revealed a diverse array of ARGs across the various drug classes, revealing both shared trends and unique features. In COVID-19 patients, 1,344 unique ARGs were identified, spanning 25 drug resistance classes, while dengue patients harbored 497 ARGs across the 24 drug classes (Fig 2a). Notably, 457 ARGs were shared between both the infections, with 847 and 40 ARGs exclusive to COVID-19 and dengue respectively. The identification of 847 and 40 ARGs exclusive to COVID-19 and dengue, respectively, was based on unfiltered detection across the study samples. By grouping ARGs into antibiotic classes, both COVID-19 and dengue patients exhibited a resistome dominated by the multiple-drug resistance genes (MDR ARGs), the genes that encoded resistance to at least two or more different drug classes. Among the resistance genes, those targeting β-lactam antibiotics, such as penam, monobactam, and penem, were particularly prominent in both the diseases. Cephalosporin resistance (10%) was highest in the COVID-19 patients, while aminoglycoside resistance (9%) dominated in the dengue patients (Fig 2a). The MDR ARGs account for a significant proportion of MDR genes—49.5% in COVID-19 and 56.5% in the dengue patients.

The most common types of MDR ARG conferred resistance to the β-lactamase family, including penam, cephalosporin, penem and monobactam common in both the groups (Fig 2b). Interestingly, the primary resistance mechanism in both the infections was antibiotic inactivation—encoded by 67.7% of ARGs in the COVID-19 and 62% in the dengue patients (Fig 2c). This shared mechanism potentially underscores the transcribed microbiome's consistent adaptive strategy when faced with antimicrobial pressures. In these resistomes, we detected resistance genes for antimicrobial drugs classified as medically important for humans, including important (e.g., Pleuromutilins), highly important (e.g., Lincosamides, Tetra-cyclines), critically important (e.g., Aminoglycosides, Macrolides) and highest priority critically important (e.g., Cephalo-sporins, Polymyxins) antimicrobials by the WHO [16]. Remarkably, resistance to non-medically authorized antibiotics like aminocoumarins and bicyclomycins was also detected, suggesting inappropriate antibiotic use in these patients.

**Abundant ARGs reflect broader drug class resistance in COVID-19, while dengue is MDR-driven, with TEM-116 as the core ARG in both infections.** To identify the most abundant ARGs in this first-of-its-kind study, we filtered genes based on abundance and prevalence across patient samples. In COVID-19 patients, 169 ARGs stood out, while dengue patients had 38 dominant ARGs (S2 Table). The COVID-19 resistome revealed genes resistant to 16 drug classes, with MDR ARGs accounting for nearly half (49%), along with aminoglycoside (18%), tetracycline (6%), cephalosporin (4%), diaminopyrimidine (3%), fluoroquinolone (3%) and macrolide (3%). In contrast, the dengue resistome showed resistance to only 7 drug classes, with MDR ARGs comprising 60% of the total, and 23% aminoglycoside drug classes. To identify the key ARGs, we analyzed the top 30 ARGs based on their presence across the patient samples, visualizing their abundance (log10 GCPM) through heatmaps—COVID-19 (Fig 2d.i) and dengue (Fig 2d.ii).

The most abundant ARGs in the COVID-19 included *kdpE* and *TEM-116*, consistently present across all the samples, while in dengue patients exhibited high levels of *kdpE*, *TEM-116*, *MexJ*, *mphA*, *AcrS*, *tet(A)*, *acrB*, *tet(C)*, and *TEM-98* across all the samples forming a distinct resistance cluster. Notably, both diseases shared *kdpE* (n = 249 in COVID-19; n = 112 in dengue) and *TEM-116* (n = 248 in COVID-19; n = 111 in dengue) as core ARGs—present across nearly all the patients and suggesting conserved mechanisms of resistance across infections. *kdpE*, a transcriptional activator within the KdpD/KdpE two-component regulatory system, plays a crucial role in bacterial virulence and intracellular survival [17]. Its widespread presence across both diseases may enhance bacterial adaptability within the host environments, potentially facilitating persistent infections. *TEM-116*, an extended-spectrum β-lactamase (ESBL), confers resistance to the third- and fourth-generation cephalosporins [18]. Interestingly, TEM-116 was consistently abundant in both infections, samples

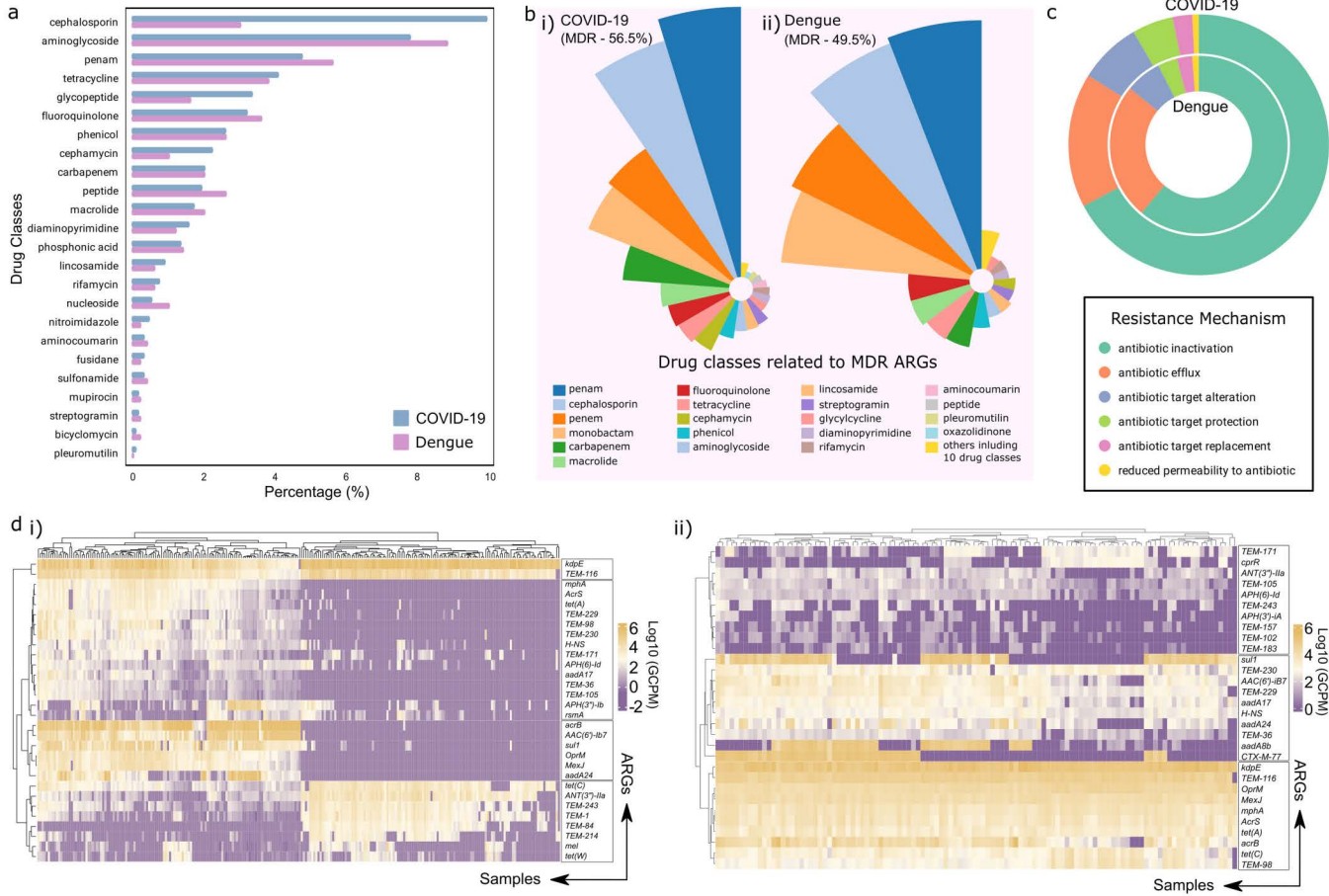

**Fig 2. The resistome profile in COVID-19 and dengue patients. (a)** Horizontal bar plot illustrating drug class abundance in COVID-19 (blue) and dengue (pink), with the y-axis representing different drug classes and the x-axis showing the percentage abundance of each drug resistance category. **(b)** Distribution of ARGs conferring multidrug resistance (MDR), highlighting similar resistance patterns in both infections. **(c)** Comparative analysis of resistance mechanisms, showing the predominance of antibiotic inactivation as the primary mechanism in both COVID-19 and dengue. **(d)** Heatmaps displaying the differential abundance (log10 GCPM) of the top 30 abundant ARGs across the patient samples in **(i)** COVID-19, and **(ii)** dengue, where purple indicates lower gene abundance and yellow represents higher gene expression levels. The highlighted boxes display ARGs clustering based on abundance patterns: the top box **(2d-i)** shows ARGs uniformly high across Dengue samples, and the bottom box **(2d-ii)** shows those high across COVID-19. Other boxes represent ARGs that are abundant in a subset of patients but low in others, illustrating inter-patient variability in resistome profiles.

collected at the point of hospital admission. As no antibiotics had been administered before this stage, the observed ARG profiles are likely indicative of baseline or community-acquired resistance, influenced by prior exposure to environmental antibiotic pressures. Its consistently high abundance in both infections suggests the presence of pre-existing reservoirs shaped by external antibiotic pressures, underscoring the role of ARGs as a potential reservoir for resistance. The additional core ARGs identified in dengue patients point to unique selective pressures shaping the resistome in this infection. Furthermore, several ARGs demonstrated high abundance in a subset of patients (shown by distinct clustering in the heatmaps) while being less prevalent in others (*acrB*, *AAC(6')-Ib7*, *sul1*, *OPrM*, *MexJ*, *aadA24* in COVID-19 and *sul1*, *TEM-230*, *aadA8b*, *CTX-M-77* in dengue), highlighting inter-patient variability in the resistome profiles (Fig 2d.i and 2d. ii). These findings reflect the intricate dynamics of resistance gene presence and the adaptive strategies employed by the microbial communities in different disease contexts.

## Taxonomic analysis of ARG hosts identifies *Escherichia coli* and *Klebsiella pneumoniae* as major reservoirs in both infections

To investigate the microbial origin of the identified ARGs, we analyzed the holo RNA-seq-derived microbiome data to determine which transcriptionally active microbes (TAMs) potentially contribute to antibiotic resistance in COVID-19 and dengue patients.

**Microbial diversity and abundance.** Our analysis identified 6,897 microbial species in COVID-19 and 5,805 in the dengue patients, both originating from 58 phyla. To determine the key contributors to the resistome, we first identified the most abundant TAMs from the total microbiome, as these species are likely to play a major role in ARG dissemination. Among them, 80 TAMs in COVID-19 and 121 TAMs in dengue were classified as abundant, defined by a relative abundance of greater than 0.1% (Fig 3a.i and 3b.i). Given their increased abundance, these TAMs represent the core microbial populations potentially driving antibiotic resistance during the viral infections. Therefore, we focused on these abundant TAMs to identify the key ARG-harboring species contributing to the overall resistome.

**Resistome-microbiome association.** Firstly, we explored the relationship between the composition of microbial communities and the abundance of ARGs in both the infections using Procrustes analysis, which revealed a significant correlation (Procrustes sum of squares $M2 = 4.441e-16$, $p = 0.001$ in COVID-19; $M2 = 8.882e-16$, $p = 0.001$ in dengue) between the composition of the resistance genes (ARGs) and bacterial composition (TAMs) (Fig 3a.ii and 3b.ii). These results suggest that a specific abundant bacterial population is closely linked to the resistome composition in both COVID-19 and dengue diseases.

**ARG-carrying TAMs.** To investigate further, we assessed the bacterial origin of the ARGs using the CARD and NCBI-NR database and identified the most abundant ARG-hosting TAMs within the transcribed microbiomes of the two patient groups. Among the core microbial population, a total of 37 TAMs in COVID-19 and 13 TAMs in dengue patients were found to carry ARGs, as highlighted in the sankey plot (Fig 3a.iii and 3b.iii). Comparing the annotated ARGs with their potential host bacteria reveals their widespread distribution across the bacterial phyla, where the maximum isProteobacteria followed by Firmicutes in both the infections. Notably, genera such as *Escherichia*, *Klebsiella*, *Acinetobacter*, *Salmonella*, *Staphylococcus*, and *Enterococcus* are prominent carriers of ARGs, with *Escherichia coli* and *Klebsiella pneumoniae* emerging as the primary hosts, carrying the highest number of ARGs in both the infections (Fig 3a.iii and 3b.iii). Whereas, *Bacillus*, *Pseudomonas* and *Streptococcus* genus harbor a higher number of ARGs in COVID-19 compared to the dengue patients. A detailed list of ARGs and their corresponding TAM host is provided in the S3 Table.

Subsequently, we identified bacterial species with a higher prevalence of ARGs for both the infections. Network analysis of co-occurrence patterns amongst the TAMs carrying ARGs revealed key species in each disease. In COVID-19 patients, the top five TAMs harboring ARGs were *K. pneumoniae*, E. *coli*, *Acinetobacter baumannii*, *Pseudomonas aeruginosa*, and *Salmonella enterica*. Meanwhile, in dengue patients, *Escherichia coli*, *Klebsiella pneumoniae*, *Salmonella enterica*, *Acinetobacter baumannii*, and *Staphylococcus aureus* were the most prevalent ARG carriers.

A total of 131 ARGs were correlated with these five bacterial species in COVID-19, whereas 22 ARGs showed a correlation in dengue. Among these, *K. pneumoniae* in the COVID-19 patients carried the highest number of ARGs (93), while *E. coli* was the dominant ARG carrier in dengue patients, with 16 genes (Fig 3a.iv). In COVID-19, 112 MDR ARGs and genes from other antibiotic classes were significantly associated with the core bacterial species. *K. pneumoniae* and *E. coli* showed the highest number of ARG correlations (63 and 60, respectively), followed by *A. baumannii* (29 ARGs), *P. aeruginosa* (26 ARGs), and *S. enterica* (20 ARGs). In contrast, dengue patients exhibited limited resistome, with only seven ARGs linked to the MDR, aminoglycoside, tetracycline, and sulfonamide resistance. Notably, *Acinetobacter* in dengue patients showed no significant correlation with any ARGs, while *E. coli* was associated with the key resistance genes

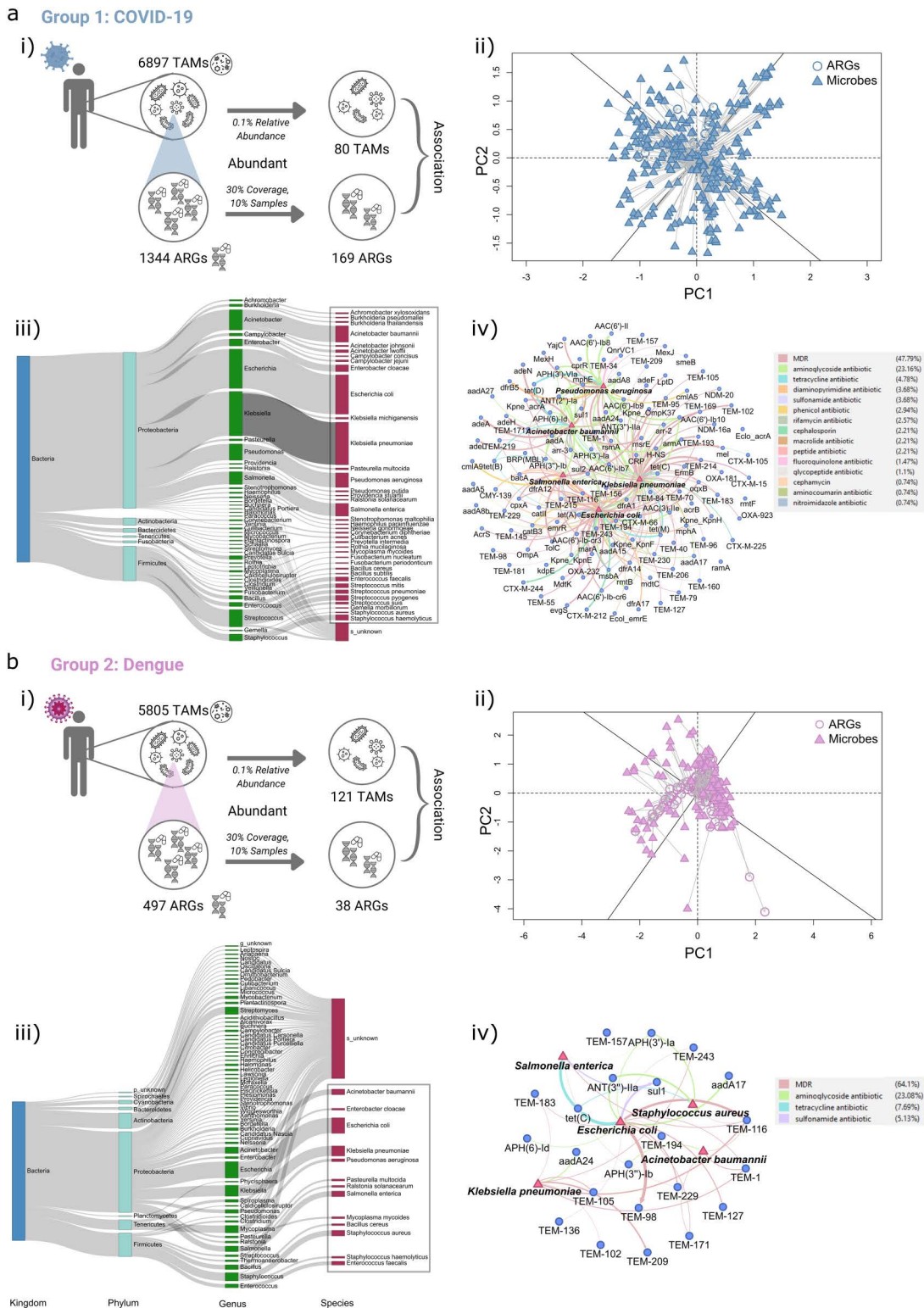

**Fig 3. Associations between ARGs and TAMs in (a) COVID-19 and (b) Dengue. (i)** Graphical representation of the numbers of ARGs and TAMs. **(ii)** Procrustes analysis of the correlations between ARGs and TAMs based on principal coordinate analysis (Bray–Curtis) results. **(iii)** Sankey plot indicating

taxonomic distribution of ARGs harboring TAMs. The colors of rectangles represent different taxonomic levels. The length of the rectangles indicates the number of ARGs. **(iv)** Co-occurrence network showing the top 5 TAMs and their associated ARGs. Circles represent ARGs and triangles represent TAMs.

such as *aadA17*, *TEM-116*, *tet(C)*, *sul1*, and *TEM-98* (Fig 3b.iv). The above findings indicate a strong connection between the ARGs and bacterial taxa in both the infections, with a markedly higher number of ARG-host associations in COVID-19 compared to dengue. This suggests a more extensive resistome profile in COVID-19 patients, potentially influenced by the differences in microbial composition and selective pressures during infection.

## Comparative analysis of microbiome and resistome profiles in COVID-19 and Dengue patients

Despite having distinct entry routes, both COVID-19 and dengue viral infections share key biological and clinical characteristics, possibly as both are caused by the single-stranded RNA viruses (SARS-CoV-2 and DENV, respectively) and pose significant public health challenges in the endemic regions. Exploring the parallels in microbial and resistome profiles between the two infections is thus crucial for identifying potential drivers of disease severity and AMR dynamics. This study compared the ARGs and microbiome compositions in the patient samples from these infections to uncover shared (commonalities) and distinct (differences) patterns. While we acknowledge that comparing clinical specimens from different body sites—nasopharyngeal swabs for COVID-19 and blood for Dengue—can introduce potential variability due to the site-specific microbiome differences, our primary aim was to characterise the TAMs and resistome at the most clinically relevant site of infection for each disease: the respiratory tract for COVID-19 and the bloodstream for Dengue. To further evaluate whether the observed microbial and genes signatures were potential disease-specific microbial rather than artifacts of sampling type, we performed a sensitivity analysis within the Dengue cohort, using an independent, lab generated dataset. and observed distinct clustering, suggesting that the observed microbial differences are attributable to disease status rather than the sampling site (S1 File). These findings reinforce the biological relevance of the identified microbial and ARGs profiles.

**Significant differences in microbial and ARG diversity between COVID-19 and dengue.** To assess the microbiome and resistome abundance and diversity between the two infections, we analyzed the meta-transcriptomics datasets from the patient samples using consistent pipeline and parameters. Alpha diversity metrics revealed notable differences between the two infections. The COVID-19 group exhibited significantly higher Shannon diversity for the microbial community than the dengue (p-value = 0.0003) (Fig 4a.i), while other metrics, such as Simpson's evenness, Chao1 abundance, and observed richness, also indicated marked differences (S2 File). This indicates a greater microbiome diversity in the COVID-19 patients. Conversely, a separate Shannon index was assessed for the ARGs and showed significantly higher alpha diversity in the dengue patients (p-value = 8.85e-12), reflecting an even distribution of the genes in the dengue group (Fig 4a.ii). Beta diversity analysis, assessed using the Bray–Curtis index, demonstrated a clear separation between the microbial communities in the COVID-19 and dengue patients (PERMANOVA; p-value = 0.001, $R^2$ = 0.1452741) (Fig 4b.i). Similarly, principal coordinate analysis (PCoA) further confirmed distinct clustering patterns of ARG profiles between the two groups, indicating significant differences in ARG abundance and composition (PERMANOVA; p-value = 0.001, $R^2$ = 0.2748598) (Fig 4b.ii).

**Taxonomic distribution of microbial communities reveal a dysbiotic, pathogen-enriched microbiome in COVID-19 compared to a more diverse and beneficial microbiome in dengue.** Both COVID-19 and dengue exhibited microbial communities dominated by four major bacterial phyla: Proteobacteria, Firmicutes, Actinobacteria, and Bacteroidetes (Fig 4c.i). Within these, genera with relative abundances exceeding 1% were commonly identified, including *Bacillus*, *Staphylococcus*, *Clostridium*, *Acinetobacter*, *Burkholderia*, and *Corynebacterium*. However, distinct

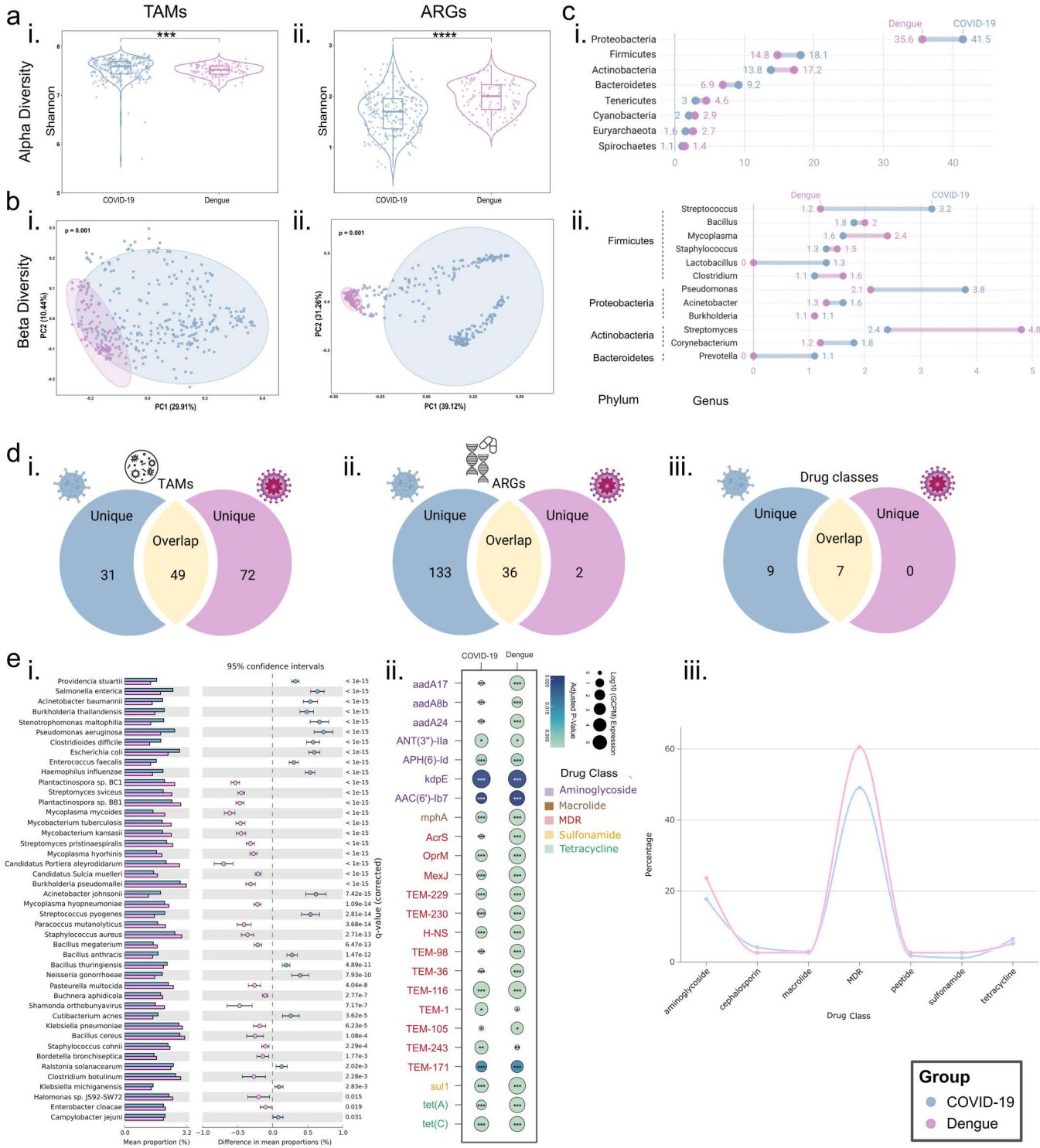

**Fig 4. Comparative resistome and microbiome diversity in COVID-19 and dengue patients. (a)** Alpha diversity indices comparing, indicating **(i)** higher microbial diversity in COVID-19, and **(ii)** higher ARGs diversity in dengue. **(b)** Beta diversity analysis, illustrating **(i)** significant differences in microbial diversity (p = 0.001), and **(ii)** distinct ARG profiles between COVID-19, and dengue (p = 0.001). **(c)** Relative abundance comparison of different microbial **(i)** phyla, and **(i)** genera. **(d)** Venn diagram illustrating unique and shared **(i)** TAMs, **(ii)** ARGs, and **(iii)** drug resistance classes in the COVID-19

and dengue. **(e)** Overlapping microbiome and resistome profiling, including **(i)** Common TAMs abundance, **(ii)** Differential expression of common ARGs, and **(iii)** Common drug class percentages in COVID-19 and dengue.

taxonomic trends emerged between the two infections, likely reflecting differences in the infection sites of SARS-CoV-2 and DENV. COVID-19 patients exhibited higher abundances of *Streptococcus* and *Pseudomonas*, genera frequently associated with respiratory infections and secondary bacterial pneumonia. In contrast, dengue patients showed an increased representation of *Mycoplasma* and *Streptomyces* (Fig 4c.ii), the latter known for its role in antibiotic production and potential immune modulation.

At the TAMs level, COVID-19 patients harbored 80 abundant TAM species, while dengue patients harbored 121 species, with 49 TAMs overlapping between the two groups (S4 Table) (Fig 4d.i). Among these shared TAMs, 44 showed significant abundance differences, with 19 species enriched in COVID-19 and 25 enriched in dengue (Fig 4e.i). These differences might be a feature of the dysbiotic nasopharyngeal microbiota dominated by the resistant pathogenic strains in COVID-19 along with the distinct infection sites of SARS-CoV-2 and dengue virus. The predominance of opportunistic and potentially drug-resistant pathogens in COVID-19, such as *A. baumannii, Acinetobacter johnsonii, Cutibacterium acnes, Enterococcus faecalis, Neisseria gonorrhoeae, P. aeruginosa, S. enterica, Stenotrophomonas maltophilia,* and *Streptococcus pyogenes*, highlights the dysbiotic nature of the COVID-19-associated microbiome. Several of these species have been linked to increased mortality and are known reservoirs of MDR ARGs, further complicating disease outcomes.

A distinct yet overlapping microbial signature was observed in dengue, where *Bacillus cereus, Burkholderia pseudomallei, K. pneumoniae, Mycobacterium tuberculosis,* and *Staphylococcus aureus* were among the prominent opportunistic pathogens. Unlike COVID-19, dengue-associated TAMs also included human-beneficial microbes, such as *Streptomyces pristinaespiralis* and *Streptomyces sviceus*. These species belong to the *Streptomyces* genus, renowned for producing antibiotics, with *S. pristinamycin* demonstrating efficacy against drug-resistant bacterial strains [19].

Further analysis of unique TAMs revealed that COVID-19 harbored 31 distinct species, predominantly consisting of human pathogens known to exacerbate infection severity. In contrast, dengue patients exhibited 72 unique TAMs, comprising a mix of opportunistic pathogens, commensals, and species with unknown roles in infections. Notably, dengue patients also harbored several viruses within their microbiome, including *Cyprinid herpesvirus 1, Glypta fumiferanae ichnovirus, Hepacivirus C,* and *Human endogenous retrovirus K*. The presence of these viral species raises intriguing questions about their potential role in dengue pathogenesis, immune modulation, and disease severity, warranting further investigation.

**COVID-19 exhibits a more diverse and abundant resistome than dengue, though both share core ARGs.** The resistome analysis revealed notable differences in the composition of ARGs between the two infections. Indicator ARGs, which are unique to each infection based on their occurrence, were identified. While both infections shared 36 common ARGs, COVID-19 patients harbored 133 unique ARGs, whereas dengue patients had only two (Fig 4d.ii). COVID-19 also displayed a broader range of ARGs spanning 16 drug classes, whereas dengue had no unique drug classes beyond those shared with COVID-19 (Fig 4d.iii).

Further analysis of abundant ARGs (≥30% coverage, ≥ 10% sample occurrence) identified 24 resistance genes associated with aminoglycoside, macrolide, multidrug resistance (MDR), sulfonamide, and tetracycline classes. Of these, 20 ARGs were enriched in dengue, while only four—two aminoglycoside (*ANT(3'')-IIa*, *kdpE*) and two MDR (*TEM-1, TEM-243*)—were more abundant in COVID-19. Notably, sulfonamide (*sul1*), tetracycline (*tet(A)*, *tet(C)*), and macrolide (*mphA*) ARGs were exclusively found in dengue (Wilcoxon test; adjusted $P < 0.05$, Fig 4e.ii).

COVID-19 patients exhibited resistance genes spanning 16 drug classes, compared to only seven in dengue. Both infections shared resistance to MDR, aminoglycosides, macrolides, tetracyclines, sulfonamides, cephalosporins, and peptides. However, COVID-19 patients harbored additional ARGs conferring resistance to glycopeptides, fluoroquinolones, rifamycins, diaminopyrimidines, cephamycins, nitroimidazoles, phenicols, nucleosides, and aminocoumarins (Fig 4e.iii).

This comparative analysis highlights both shared and distinct features of the resistome and microbiome in the COVID-19 and dengue. While both groups share a subset of common microbial phyla and ARGs, COVID-19 exhibits relatively extensive resistome with greater ARG diversity and abundance. The microbial community structure in dengue exhibits a range of diverse TAMs, whereas COVID-19 patients are more dominated by the opportunistic taxa. These findings underscore the importance of understanding microbial and resistome dynamics across different infections to guide antimicrobial resistance management and therapeutic strategies.

## Disease phenotypic differences explained by the ARGs and associated TAMs

To gain deeper insights into the impact and potential role of ARGs and TAMs on the differential disease severity, we analyzed patient data from both COVID-19 and dengue cohorts, stratified by severity levels: mild, moderate, severe, and mortality.

**ARG distribution and abundance across COVID-19 and dengue severity subgroups reveal increased diversity, underlie mortality in COVID-19.** In COVID-19 patients, ARG prevalence increased with severity—114 in mild, 89 in moderate, 182 in severe, and 286 in mortality cases (S5 Table). Interestingly, 53 ARGs were common across all the subgroups, while severe and mortality patients harbored a significantly higher number of unique ARGs (39 and 155, respectively) than the mild (4) and moderate (3). Conversely, dengue showed a stable ARG distribution (43 in mild, 42 in moderate, 49 in severe), with 34 core ARGs shared across the subgroups (Fig 5a.i, ii). However, certain beta-lactamase family ARGs were uniquely distributed: mild contained *TEM-55*, *TEM-232*, *TEM-149*, and *TEM-192*; moderate had *TEM-185*, *OXA-923*, and *TEM-317*; while severe patients exhibited *TEM-9*, *TEM-181*, *TEM-84*, *TEM-150*, *TEM-224*, *TEM-112*, *PmrF*, *TEM-128*, and *SHV-7*. In the context of COVID-19, the presence of comorbidies prompted an investigation into potential associations between comorbidities and antimicrobial resistance genes (ARGs) across varying disease severities. Statistical analyses, including logistic regression models for each group (mild, moderate, severe, and mortality), revealed no significant associations between these comorbidities and the prevalence of ARGs across disease severity, except in the mild group. In the mild group, a few genes showed significant but weak associations with certain comorbidities; however, these genes were not part of prevalent genes (TEM-55, TEM-232, TEM-149, and TEM-192) of COVID-19 mild which we have mentioned above, indicating that comorbidities likely did not significantly influence ARG patterns in this cohort (S6 Table). Unlike COVID-19, where ARG numbers varied significantly by severity, dengue ARG distribution remained relatively stable, suggesting a more pronounced role of ARGs in modulating COVID-19 severity.

A breakdown of drug resistance profiles revealed that MDR genes were the most prevalent in both the COVID-19 and dengue patients, followed by aminoglycoside resistance genes. Notably, severe and mortality COVID-19 cases exhibited resistance to a broader range of drug classes, including phosphonic acid, nucleoside, cephalosporin, phenicol, and aminocoumarin—absent in mild and moderate cases. Similarly, Cephalosporin, Phenicol, and Aminocoumarin resistance genes, absent in the moderate, appeared in the severe and mortality patients. Among these, cephalosporin resistance genes contributed 2% in severe patients and 6% in mortality. Dengue, on the other hand, displayed a stable drug class distribution, with MDR (~60%), aminoglycoside (~20%), and tetracycline (~4%) genes dominating across all the severity subgroups (Fig 5b.i, ii). This consistency reinforces the idea that ARGs play a crucial role in COVID-19 severity, While this could be influenced by prior antibiotic use, as mentioned before our samples were collected on the first day of hospital reporting, reducing the likelihood that elevated ARG levels are solely due to in-hospital antibiotic administration. However, increased antibiotic use before hospital admission may still be a contributing factor.

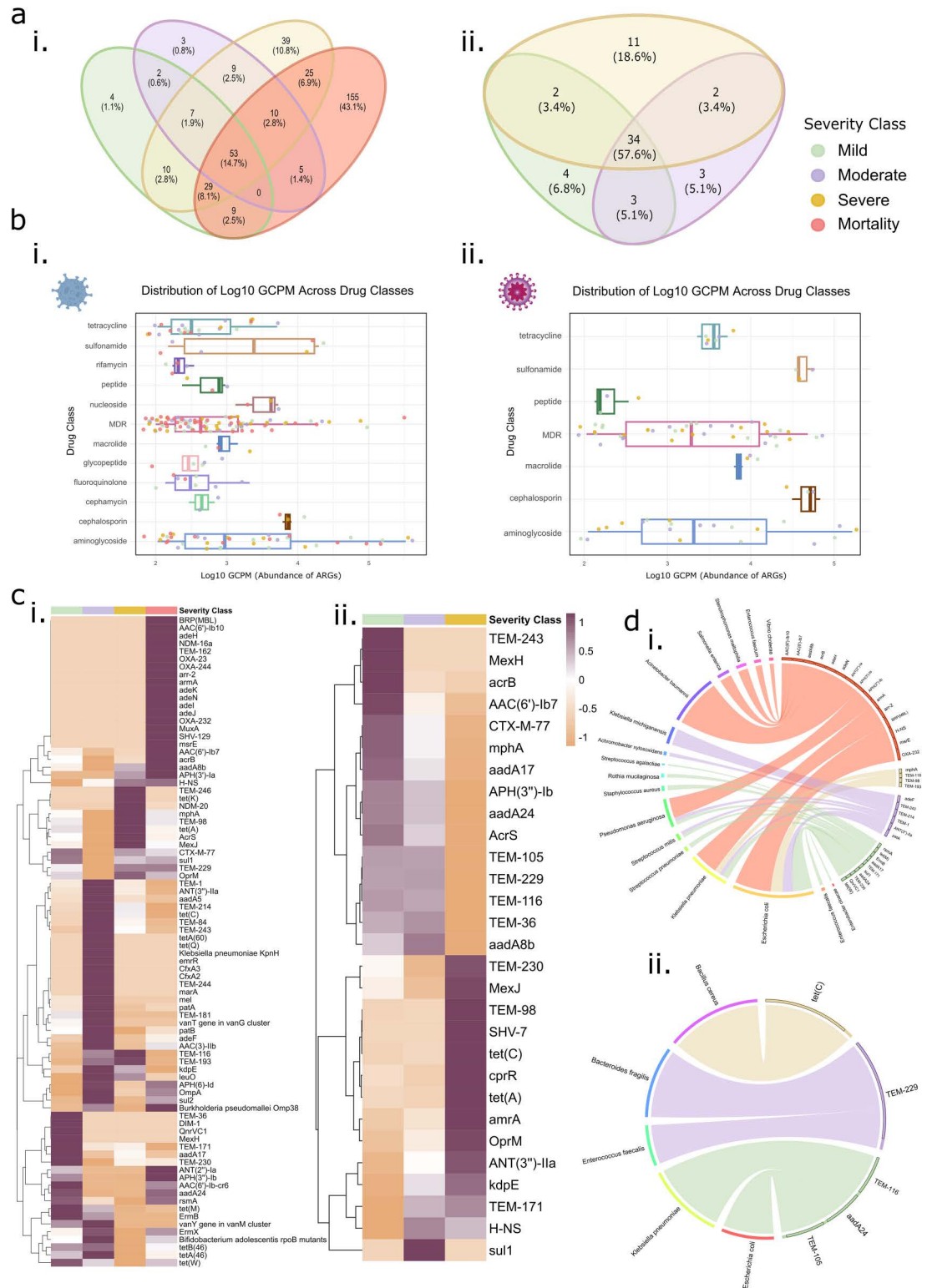

**Fig 5. ARGs distribution, drug class abundance and microbial associations across COVID-19 (i) and dengue (ii) severity groups. (a)** Venn diagram illustrating the distribution of antimicrobial resistance genes (ARGs) across the severity groups in **(i)** COVID-19, and **(ii)** Dengue, highlighting shared and unique ARGs in each infection. **(b)** Distribution of different drug resistance classes, measured as Log10 GCPM, in **(i)** COVID-19, and **(ii)**

Dengue, showcasing variations in the drug resistance gene expression across the severity levels. **(c)** Heatmap representing the abundance of ARGs (average GCPM > 100) across severity groups in **(i)** COVID-19, and **(ii)** Dengue, illustrating differential resistance gene expression patterns. **(d)** Microbial communities associated with ARGs across severity levels in **(i)** COVID-19, and **(ii)** Dengue, identifying key transcriptionally active bacterial species linked to resistance profiles.

To refine our analysis, we filtered ARGs based on their abundance (average GCPM > 100), identifying 84 highly abundant ARGs in COVID-19 and 29 in dengue. In COVID-19, the distribution of abundant ARGs varied by severity, with 50 in mild, 44 in moderate, 37 in severe, and 64 in mortality cases. While most ARGs were shared across severity groups, mortality patients harbored 18 unique ARGs, compared to 4 in mild, 5 in moderate, and 2 in severe cases. The highest average ARG abundance was observed in moderate COVID-19 patients (32 ARGs), followed by mortality (26), mild (15), and severe (11) (Fig 5c.i). Aminoglycoside and MDR ARGs were dominant across all the severity groups, except in the severe cases, where tetracycline resistance genes were more prevalent. Notably, mortality patients harbored 26 enriched ARGs, with 17 exclusive to this group, including MDR (*adeJ*, *OXA-232*, *MuxA*, *SHV-129*), rifamycin (*arr-2*), aminoglycoside (*AAC(6')-Ib10*, *armA*), and glycopeptide (*BRP(MBL)*) resistance genes.

In dengue, ARG distribution remained relatively stable across the severity subgroups, but notable shifts were observed. Severe cases had 12 highly abundant ARGs, while mild cases exhibited a cluster of 12 enriched ARGs, which decreased in severe cases (Fig 5c.ii). Moderate cases showed a distinct increase in *aadA8b*, *sul1*, and MDR-associated ARGs (*TEM-36*, *H-NS*, *TEM-229*). Mild cases were enriched in aminoglycoside ARGs (*APH(3'')-Ib*, *AAC(6')-Ib7*, *aadA24*) and a few MDR genes (*TEM-243*, *MexH*, *acrB*). The *kdpE* gene was consistently abundant across all the severity groups but was significantly higher in severe cases, suggesting its role in increased microbial virulence. Additionally, severe dengue cases showed elevated tetracycline resistance genes (*tet(A)*, *tet(C)*) along with aminoglycoside (*ANT(3'')-IIa*) and MDR ARGs (*amrA*, *TEM-171*, *TEM-230*, *TEM-98*, *MexJ*, *OprM*, *SHV-7*), as well as a single peptide resistance gene (*cprR*). These patterns indicate that while COVID-19 mortality cases exhibit a broader resistome with high MDR enrichment, dengue severity is associated with specific ARG shifts, particularly in tetracycline and aminoglycoside resistance.

**ARGs - host TAMs correlations: Stronger ARG-TAM associations in COVID-19 mortality suggest a key role of bacterial reservoirs.** When examining the bacterial hosts (TAMs) associated with ARGs, we identified that the COVID-19 patients exhibited a higher number of ARGs-TAMs correlations across all the severity subgroups compared to the dengue. The number of ARGs and TAMs identified in each severity group is summarized in the table below (Table 2):

Subsequently, to investigate severity-specific TAMs harboring ARGs, we identified the most abundant ARGs in each severity group and traced their bacterial hosts. The resulting significant ARG-TAM correlations are illustrated in the circos plot (Fig 5d.i for COVID-19 and 5d.ii for dengue). In COVID-19, *E. coli* and *K. pneumoniae* were strongly associated with ARGs across all severity levels. While moderate COVID-19 cases exhibited the highest number of abundant ARGs, mortality cases displayed the most extensive ARG-TAM correlations, with 10 ARGs linked to 10 bacterial species—*A. baumannii* and *E. coli* being the predominant contributors (Fig 5d.i). Additionally, each severity group exhibited unique TAM signatures harboring ARGs (Table 3). Conversely, dengue patients exhibited distinct microbial associations across severity

**Table 2. Number of ARGs and TAMs in differential severity groups of COVID-19 and Dengue.**

| Severity | COVID-19 (*ARGs*) | COVID-19 (*TAMs*) | Dengue (*ARGs*) | Dengue (*TAMs*) |
|---|---|---|---|---|
| Mild | 79 | 29 | 20 | 9 |
| Moderate | 40 | 20 | 22 | 11 |
| Severe | 63 | 20 | 17 | 9 |
| Mortality | 122 | 28 | – | – |

**Table 3. Unique TAMs harboring ARGs in COVID-19 and Dengue.**

| Severity | COVID-19 TAMs | Dengue TAMs |
|---|---|---|
| Mild | *E. cloacae, E. faecalis, S. pneumoniae, S. aureus, R. mucilaginosa, S. agalactiae* | *K. pneumoniae* |
| Moderate | *A. xylosoxidans, K. michiganensis* | *B. fragilis, E. faecalis* |
| Severe | – | *B. cereus* |
| Mortality | *A. baumannii, S. enterica, S. maltophilia, E. faecium, V. cholerae* | – |

groups. In mild cases, *E. coli* and *K. pneumoniae* were the primary ARG-harboring TAMs, while *E. faecalis* and *B. fragilis* were predominant in the moderate. Notably, in severe cases, *B. cereus* was the sole TAM, carrying tetracycline resistance gene (*tet(C)*) (Fig 5d.ii).

COVID-19 patients showed stronger correlation with a broader range of *ARGs* from the diverse drug classes, with the highest representation in the mortality patients. In contrast, dengue severity subgroups exhibited a stable ARG and drug class profile. While moderate and severe dengue cases showed ARG-TAM correlations primarily with tetracycline (*tet(C)*) and MDR genes, mild patients also had correlations with sulfonamide (*sul1*) and aminoglycoside (*aadA24*).

These findings suggest a plausible link between the ARG diversity and COVID-19 severity, while dengue severity groups remain more stable in terms of ARG and drug class profiles. The presence of cephalosporin and phosphonic acid resistance genes in the severe and mortality COVID-19 patients suggests their potential role in disease progression. Additionally, the differential abundance of ARGs and their association with specific TAMs highlight the significance of bacterial reservoirs in shaping ARG dynamics in both the infections.

To validate and extend our findings, we analyzed data from an independent in-house cohort of 58 dengue patients (26 severe, 32 non-severe; classified per WHO guidelines), along with dengue-negative febrile controls (NS1-negative). We identified 300 ARGs across the discovery (n = 112) and validation (n = 58) cohorts, with 208 shared between them. Using the same thresholds—≥ 30% coverage in ≥10% of samples for ARGs and ≥0.1% relative abundance in ≥50% of samples for TAMs—we observed strong concordance between cohorts. ARG abundance was highly correlated ($R^2 = 0.93$, p = 0.0067), as was microbial composition ($R^2 = 0.77$, $p < 2.2 \times 10^{-16}$), supporting the robustness and reproducibility of our approach across independent patient groups (S3 File).

## Discussion

COVID-19 and dengue have caused widespread outbreaks across Asia and Latin America, with overlapping clinical presentations, including fever, headache, myalgia, and gastrointestinal symptoms. Despite their distinct entry routes, both SARS-CoV-2 and dengue virus (DENV) cause systemic illness. The absence of specific antiviral treatments or vaccines for the disease has led to the widespread use—and potential misuse—of antibiotics, particularly during the peak of the COVID-19 pandemic. A similar concern exists for dengue fever, where empirical antibiotic use remains common despite its viral etiology. While the increasing adoption of genomics-based surveillance has enhanced AMR monitoring, previous research on antimicrobial resistance genes (ARGs) in COVID-19 and dengue infections has primarily focused on phenotypic AMR surveillance [20,21]. To bridge this gap, we employed non-canonical meta-transcriptomic sequencing, allowing us to explore the distribution and functional activity of ARGs within the TAMs in the infected patients, using a suite of bioinformatics strategies. To our knowledge, no comparative, unbiased genomics-based study has systematically examined the resistome of these two single-stranded RNA viral infections. Here, we provide unexplored insights into the differential resistome landscapes of COVID-19 and dengue, revealing distinct shifts in ARG composition and host bacterial profiles across the disease severity subgroups. These findings highlight the urgent need to curb inappropriate antibiotic use in viral infections, as the emergence of AMR in co-infecting bacterial populations may exacerbate disease severity and complicate treatment strategies.

COVID-19 patients exhibited a higher diversity of ARGs, suggesting increased antibiotic misuse during the pandemic in driving antimicrobial resistance. Our findings reveal that the prevalence of multidrug-resistant (MDR) ARGs is notably high in both the infections, with MDR β-lactamases as the most prevalent MDR mechanism encountered in the clinical settings. Yet, subtle distinctions could be noted between the MDR patterns in COVID-19 and dengue. Along with *TEM* (*ESBL*), carbapenemase resistance ARGs, *NDM*, *OXA*, *KPC* and *VIM* dominated COVID-19, whereas *TEM* alone was majorly associated with dengue. Similarly, other *ESBLs*, *CTX-M* and *SHV* (providing resistance against cephalosporin drug class), exhibited higher presence in COVID-19 than dengue. Use of broad-spectrum antibiotics, due to its presumed effect on SARS-CoV-2 as well as combating secondary infections, led to selection pressure and resistance emergence in co-infecting or co-colonising pathogens, increasing the diversity of ARGs in COVID-19 [6]. A study indicated that up to 72% of COVID-19 patients received cephalosporins in their treatment regimen [22]. Contrarily, dengue patients generally do not receive aggressive antibiotic therapy, owing to lower rates of incidence of co-infection, thus presenting a conserved MDR resistome with fewer ARGs. These findings align with previous research indicating a rise in antimicrobial resistance following the COVID-19 pandemic [23]. Interestingly, our study identified a unique ARG associated with the pleuromutilin drug class in the COVID-19 patients, which was not detected in the microbiome of the Dengue patients. Pleuromutilin, specifically lefamulin, has been shown to be effective against community-acquired bacterial pneumonia (CABP) in humans [24]. Although there is currently no evidence of pleuromutilin being used in COVID-19 cases, its efficacy against respiratory tract infections raises the possibility of its future application in managing secondary bacterial infections in the COVID-19 patients or similar diseases. Interestingly, in our study, mild cases of COVID-19 exhibited the highest antibiotic exposure, with macrolides (22.7%), tetracyclines (14.1%), and beta-lactams (3.9%) being the most frequently prescribed. In contrast, severe and mortality groups showed lower overall antibiotic usage. Macrolides were administered to 20.8% of severe cases and 6.7% of mortality cases. Tetracyclines were more used in mild patients followed by severe (8.3%), moderate (6.1%) and mortality (0). Our clinical data indicate that β-lactam antibiotics were administered during treatment at varying frequencies (e.g., 3.9% in mild, 9.1 in moderate, 12.5% in severe, and 0% in mortality groups). While these data provide valuable insights into treatment patterns, they hold limited relevance to the ARG profiles detected, given the temporal disconnect between treatment and sample collection. It was worth noting that no patients in the mortality group received β-lactam and tetracyclines antibiotics. Despite the higher antibiotic exposure in the mild and moderate groups, resistance genes associated with cephalosporins, phenicols, and aminocoumarins, were predominantly detected in the severe and mortality groups. This apparent discrepancy highlights the complexity of ARG dynamics, suggesting that factors such as baseline resistance profiles, microbial ecology, and environmental exposures may play significant roles in ARG prevalence in severe cases.

Previous literature had reported that COVID-19 and dengue co-infection had worse outcomes regarding mortality rates, ICU admission, and prolonged hospital stay [25]. Thus, it is important to take wise-decision and clinical management approaches for the patients to enhance their clinical outcomes. Establishing an early diagnosis might be the answer to reducing the estimated significant burden of these diseases. In our study, we compared the microbial flora giving rise to the ARG reservoir in the clinical samples from patients with SARS-CoV-2 and DENV infections. The COVID-19 microbiome (nasopharyngeal region) was more diverse driven by respiratory dysbiosis, with more opportunistic pathogens acquiring resistance genes whereas dengue microbiome was more balanced likely due to restricted microbial entry into the bloodstream from the gut or other reservoirs. While both infections shared opportunistic TAMs, yet the association of the ARGs with TAMs was substantially distinct. COVID-19 presented with many opportunistic pathogens harboring the ARGs, *K. pneumoniae*, *A. baumannii*, *P. aeruginosa*, and *S. enterica* along with *E. coli*. A similar finding has been observed and reported in a systematic review of AMR in COVID-19 patients [26]. The upper respiratory tract harbors a diverse microbial community, increasing the likelihood of horizontal gene transfer (HGT) of ARGs between the species. Studies have reported increased resistance in pathogens such as *A. baumannii* and *K. pneumoniae* during the COVID-19 period [27]. *K pneumoniae* and *E coli* showed association with CTX-M gene, which is a major cause of Cephalosporin resistance

observed in COVID-19. Similarly, *K pneumoniae* and *E coli* are the major hubs for MDR. MDR β-lactamases, particularly carbapenem-resistant Enterobacterales (CRE) and extended-spectrum β-lactamases (*ESBLs*), form the most common MDR mechanisms limiting antimicrobial options [28]. In dengue, MDR pathogens like *K. pneumoniae* and *E. coli* dominate. Blood microbiome might involve gut-translocated *E. coli* and *K. pneumoniae*, which frequently carry *TEM* (*ESBL*), as observed in our study. It is important to point out that diet is a major modulator of the gut microbiome, however, its direct impact on the blood and nasal microbiomes (based on the studies diseases) is minimal and largely indirect. Unlike the gut, there is limited evidence linking dietary patterns to microbial composition in these sites [29]. Most dietary influences on host health are mediated through gut-derived metabolites that circulate systemically, rather than through direct microbial colonization of blood or nasal tissues [30,31]. Since our study focuses on microbiomes at the primary sites of infection—nasopharyngeal swabs for COVID-19 and blood for Dengue—dietary effects are unlikely to significantly confound our findings, though a minor influence cannot be entirely excluded.

A question persists as to why same/similar pathogen infections lead to differential disease severity and clinical outcome? Towards that, uniquely, our research findings provide significant insight into the distribution and composition of ARGs across the severity classes (mild, moderate, severe and mortality), particularly in the COVID-19 patients, where we observed a marked increase in ARG prevalence in the severe and mortality cases. A systematic review previously reported that nearly all mortality patients were infected with resistant organisms [26]. In our study, MDR, aminoglycoside, glycopeptide, rifamycin, and other drug resistance genes were enriched in the COVID-19 mortality patients, with various TAMs serving as reservoirs. Notably, *Acinetobacter baumannii* emerged as a key contributor to aminoglycoside and MDR resistance genes in the COVID-19 mortality group. This aligns with previous reports identifying *A. baumannii* as a major causative agent of respiratory infections and bacteremia in critically ill COVID-19 patients, often leading to outbreaks in the intensive care units [32]. Conversely, in dengue, ARG profiles remained relatively stable across the severity subgroups. The presence of *Bacillus cereus* as an ARG reservoir in severe dengue cases corroborates our earlier findings on dengue-associated TAMs, where *B. cereus* was identified in severe cases (Fig 6) [12]. This microbe has been studied for its toxin gene profile and antibiotic resistance, further supporting its role in severe dengue pathophysiology [33]. While antibiotic use in dengue patients remains moderate, inappropriate usage has been documented, underscoring the necessity for a clearer understanding of AMR dynamics in viral infections.

## Conclusion

Although our study is cross-sectional rather than longitudinal, it provides a comprehensive snapshot of the resistome landscape across the COVID-19 and dengue severity classifications. Longitudinal data can assess temporal changes in AMR dynamics. We identified *E. coli* and *K. pneumoniae* as the dominant ARG-harboring bacterial species in both infections and across all severity levels, suggesting a shared AMR reservoir between these two viral diseases. To summarize, this is the first study to leverage holo RNA-seq data with non-canonical approach to characterize the resistome of the two major infections, COVID-19 and dengue, unveiling the presence of numerous genes conferring resistance to 24/25 different drug classes. The findings emphasize the importance of understanding the comprehensive picture of the resistome and their host bacterial species and to utilize this information to develop concrete antibiotic resistance mitigation strategies. These strategies can strengthen the judicious use of antibiotics and guide the selection of specific treatments for viral diseases, COVID-19 and dengue. The identification of both unique and shared ARGs between these diseases underscores the complex interaction between viral infections and microbial resistance, highlighting the need for further research to explore the impact of these dynamics on patient care and public health. In COVID-19 patients, ARGs were not only more numerous but also more abundant, as reflected in gene copy per million (GCPM) values. This increased burden was largely driven by opportunistic bacterial species, including *K. pneumoniae, E. coli, A. baumannii, P. aeruginosa,* and *S. enterica.* However, despite these findings, key aspects of the resistome and microbiome in COVID-19 and dengue remain poorly understood, particularly regarding the extent of their overlap and divergence.

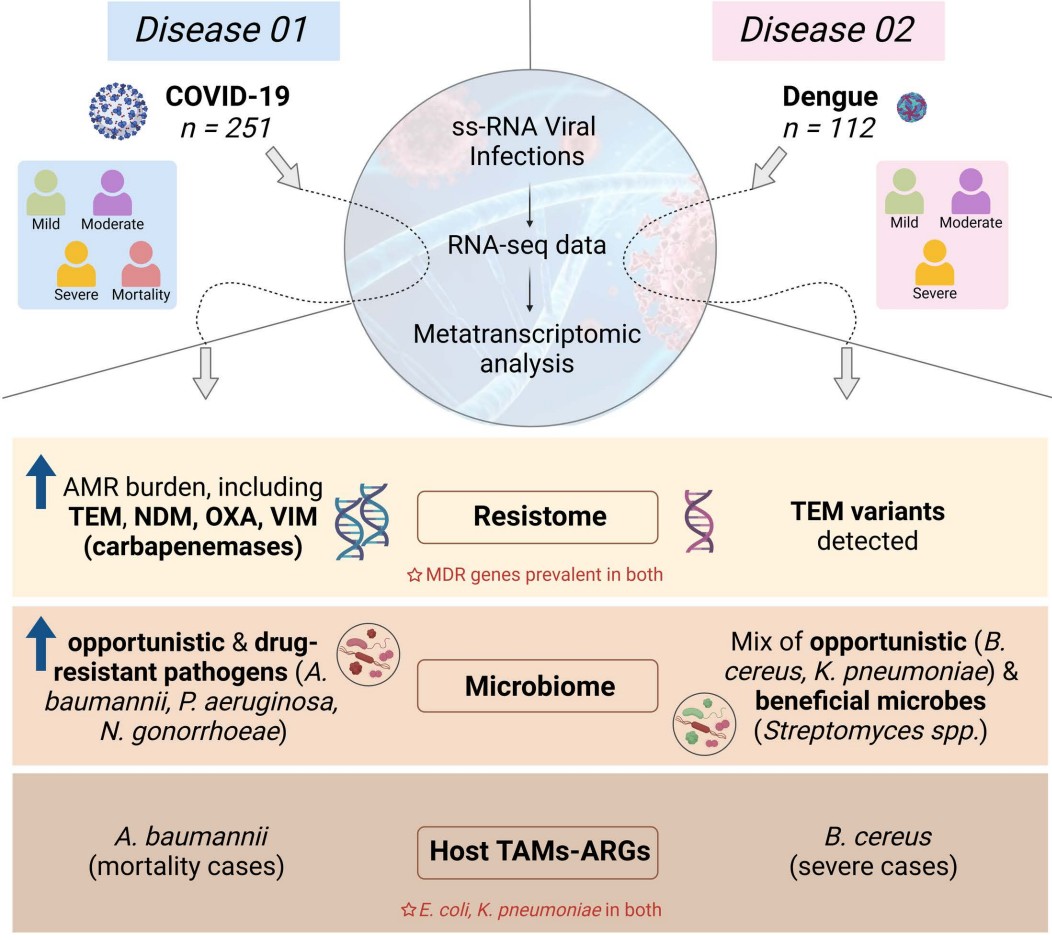

**Fig 6. Graphical representation of the key study findings.** It shows ARGs and TAMs findings across the two RNA viral infections—COVID-19 and Dengue—highlighting shared and distinct microbial and resistance profiles. *Created in BioRender. Devi, P. (2025).*

## Limitations

While our study provides detailed and important insights into the resistome and active microbial communities in the COVID-19 and dengue patients, there are a few limitations that should be addressed in future work to enhance interpretability and clinical relevance. First, the lack of detailed antibiotic usage data prior to hospital admission restricts our ability to fully correlate resistome patterns with prior antimicrobial exposure. Antibiotic usage data were only partially available for COVID-19 patients and entirely unavailable for the dengue cohort, alongside missing information on hospitalization duration and comorbidities—factors that could influence microbial signatures. Another potential limitation is the absence of dietary information, which could be a confounding factor; however, current evidence indicates that diet has minimal direct influence on the blood and nasal microbiomes. Additionally, meta-transcriptomics, while powerful in capturing active microbial and functional gene expression, reflects RNA-level dynamics, which may not always correspond to downstream protein activity. Nonetheless, transcriptomic profiles serve as a reliable surrogate for assessing real-time microbial functional dynamics, particularly in the absence of protein-level data. Future integration of complementary methods such as metaproteomics could further refine functional interpretations, but the current findings already provide meaningful and biologically relevant insights. We also acknowledge the absence of healthy controls and extraction-negative controls in the

current design. However, our objective was to compare disease-specific microbiome and resistome profiles between two infectious diseases. We plan to incorporate both extraction blanks and dengue-negative febrile illness controls in future work that will expand on this study by exploring severe and non-severe dengue cases. Finally, all patient samples in this study were collected from North India, which may limit the generalizability of our findings to other geographic regions with differing healthcare infrastructure and antibiotic usage patterns. Moreover, while this meta-transcriptomic approach offers valuable insights, its broader adoption in public health surveillance remains constrained by cost, complexity, and infrastructure—particularly in low- and middle-income countries (LMICs). Streamlining protocols and developing targeted transcriptomic panels may help increase feasibility for routine surveillance and outbreak preparedness. Further research is needed to unravel these complexities and their implications for AMR management in viral infections.

## Methods

### Ethics statement

All study procedures were in accordance with the Declaration of Helsinki. The human participants (COVID-19 and dengue patients) involved in the studies were reviewed and approved by the Institutional Ethics Committee of the CSIR-Institute of Genomics and Integrative Biology (CSIR-IGIB) Clearance (Ref No: CSIR-IGIB/IHEC/2020–21/01) and the MAX Super Specialty Healthcare Hospital, New Delhi, India. Formal verbal consent was obtained from all individuals or their legal guardians, in case of child participants.

### Patient cohort, sample collection, RNA isolation, and data acquisition

**COVID-19 patients.** The study was conducted with 363 samples, comprising 251 COVID-19 and 112 dengue cases. COVID-19 samples were collected from hospitalized patients at the MAX Super Specialty Healthcare Hospital, New Delhi, India, between April 2020 and June 2022. Nasopharyngeal swab samples were collected in Viral Transport Medium (VTM) (HiViral Transport Kit, HiMedia, Cat. No: MS2760A-50NO) by the trained paramedical staff as per the standard Indian Council of Medical Research (ICMR) guidelines. Samples were vortexed and centrifuged followed by RNA extraction using the QIAmp viral mini kit, Qiagen.

Alongside, the detailed hospital-registered clinical data of the patients were electronically recorded. The patients were further sub-grouped into mild, moderate and severe categories. Briefly, $SpO_2$ levels, requirement of respiratory support, and breathlessness parameters were taken into consideration. The $SpO_2$ level was ≥ 94% in the mild patients with no breathing problems. Moderate patients showed breathing difficulty with $SpO_2$ levels ranging between 91–93%. Severe patients showed respiratory distress with respiratory support requirement and $SpO_2$ levels < 90%. The mortality group was defined as patients who succumbed to COVID-19 during their hospital stay.

**Dengue patients.** To investigate dengue cases, 112 blood samples were collected from the dengue-positive individuals (confirmed via NS1 antigen testing) by the medical team at the MAX Super Specialty Healthcare Hospital, New Delhi, India, during the patient's initial hospital visit. This sample procurement phase spanned from August to November 2022, coinciding with the peak dengue transmission period in New Delhi, India. Pertinent clinical data was subsequently retrieved from the electronic health records (EHR) of the study participants, categorizing the 112 patients into two groups: without warning signs (mild; n = 46) - no leukopenia or thrombocytopenia and with warning signs (moderate; n = 45, leukopenia but no thrombocytopenia, and severe; n = 21, both leukopenia and thrombocytopenia). Notably, none of the individuals displayed severe bleeding, organ failure, or abnormal liver function.

RNA extraction from these samples was conducted using the QIAGEN QIAamp RNA Blood Mini Kit with specific protocol modifications to optimize yield and quality for library preparation. Incubation and centrifugation durations during the erythrocyte lysis step were reduced by 5 minutes to minimize RNA degradation and prevent excessive cell disruption, which can release contaminants. Additionally, a 2–3 minutes incubation was introduced during all washing steps to

enhance RNA purity by allowing more effective removal of residual proteins, salts, and other impurities that can interfere with library preparation and sequencing. RNA purity was assessed using NanoDrop and verified by agarose gel electrophoresis. Extracted RNA was stored at −80°C before library preparation.

## Library preparation and sequencing

The transcriptome sequencing for COVID-19 was performed using the Illumina TruSeq Stranded Total RNA Library Prep Gold (Illumina, Cat. No. 20020598). Before proceeding with cDNA synthesis for downstream library preparation, the depletion of cytoplasmic rRNA was carried out with 250 ng of input total RNA from each sample. Contrarily, dengue RNA sequencing libraries were prepared using Illumina TruSeq Stranded Total RNA Library Prep Globin Kit (Illumina, Cat. No. 20020612) from a total of 250 ng RNA, to account for the blood source of the RNA. This process entailed the depletion of globin mRNA and ribosomal RNA, followed by the initiation of first-strand cDNA synthesis. For both the infection samples, second-strand synthesis was then carried out using DNA polymerase I and RNase H. The resulting double-stranded cDNA was purified using AMPure XP beads (Beckman Coulter, A63881). Next, the purified cDNA underwent 3' adenylation, followed by the adapter ligation and enrichment through PCR amplification. The final libraries were purified, and quality assessment was performed using the Agilent 2100 Bioanalyzer. A final loading concentration of 650 pM was used for sequencing on the NextSeq 2000, with a paired-end read length of 2 × 151 bps [8,12].

## Sequencing data processing and meta-transcriptomic analysis

The generated paired-end reads in base calling (.bcl) format were demultiplexed using the bcltofastq (v1.8.4) and processed into the FASTQ format for downstream analysis. Quality assessment was performed using FastQC (v0.11.9), and the adapter content and low-quality reads were filtered using Trimmomatic (v0.39). Human genome sequences were removed by aligning reads to the reference (Ensembl hg38) using HISAT2 (v2.2.1). Non-human reads taxonomical classification was performed using Kraken2 (v2.1.3) with the minikraken2 database (v1, 03/2020), which consists of bacteria, archaea, and viral complete sequences from RefSeq. Quantification of reads and relative abundance of microbial species was characterised using Bracken (v2.9). Additionally, to confirm the presence of primary pathogens, sequencing reads were mapped to the reference genomes of Dengue virus and SARS-CoV-2. Dengue virus reads were consistently detected in all 112 dengue-positive samples, with genome coverage ranging from 17% to 100%. In the COVID-19 cohort, SARS-CoV-2 reads were detected in 213 samples, with coverage ranging from 20% to 100% (S4 File). Sample-wise microbial Abundance tables were merged using "combine_bracken_outputs.py", the combined data was normalised with the CSS (cumulative sum scaling) method using the "cumNorm" function of "metagenomeSeq" Package (v1.46.0) in R Software (v4.4.1). Although Bracken was originally designed for DNA-based metagenomic data, we used the Kraken2-Bracken pipeline due to its proven accuracy and robustness, even with low-quality or complex read datasets. Benchmarking studies have validated Kraken2/Bracken's high classification accuracy and its ability to refine species-level abundance estimates by probabilistically reallocating reads from higher taxonomic levels [34,35]. To ensure the robustness of our analysis, we evaluated multiple approaches for calculating relative abundance—including direct use of Bracken raw and normalized counts, and manual normalization. Across all methods, the microbial distribution patterns and top taxa remained consistent, indicating that the relative abundance estimates are stable regardless of the calculation strategy (S5 File). Further, the biom convert function of the "biom" package (v2.1.7) was used to convert the abundance table in biom format which was then used to calculate Alpha and beta diversity. Alpha diversity indices of all samples were calculated using the R function "estimate_richness" and Bray–Curtis dissimilarity (Beta diversity) values were calculated for all samples among two conditions using the R function "vegdist" in the "vegan" package (v2.6-8). PCoA "phyloseq" package (v1.38.0) was used to ordinate each dissimilarity matrix and visualized as a pCoA plot. Further, to identify the most abundant species, TAMs were selected based on relative abundance thresholds (>0.1% for species and >1% for genera and phyla). The >0.1% relative abundance cutoff is commonly used in microbiome studies to balance sensitivity

while minimizing noise from low-abundance taxa [36–38]. This threshold also aligns with our objective to capture species that make a meaningful contribution within their respective phyla, even at modest absolute abundance. To validate this, we systematically evaluated the cumulative contribution of >0.1% abundant species to the total abundance of their corresponding phyla (S6 File). A detailed description of the experimental and analytical pipeline is available in our published STAR Protocol, ensuring full reproducibility of the methods used [39].

## ARGs and bacterial host analysis

The non-human reads were aligned to the Comprehensive Antibiotic Resistance Databases (CARD, v3.3.0) (updated 27/08/2024, accessed on 11'24) to detect the ARGs using Resistance gene identifiers "rgi" tool (v6.0.3) command rgi coupled with argument bwt and kmer based kma algorithm capturing total ARGs. For downstream analyses, ARGs were filtered based on the threshold of more than 30% coverage and presence in at least 10% of the samples of each study group. This threshold balances sensitivity and specificity, accounting for RNA-seq data fragmentation and expression variability. We stratified ARGs into 30–50% and >50% coverage groups, observing that most high-coverage ARGs were also detected in the lower range, indicating consistent detection. Dengue samples showed stable ARG counts across 30–90% coverage, while COVID-19 had several ARGs detected only below 50%. To enhance resolution, we have now provided a detailed breakdown of ARGs surpassing coverage thresholds from 30% to 90%, along with a graphical summary in the S7 File. ARG gene coverage per million (GCPM) values were calculated using the formula:

$$GCPM(x) = \frac{\frac{counts(x)}{gene\ length(x)} \times 10^6}{\sum^n \frac{counts}{gene\ length}}$$

This formula normalizes sequencing depth and gene length to quantify gene abundance. Here, GCPM(x) represents the GCPM value of gene x, counts(x) refers to the number of mapped reads for gene x, gene length(x) denotes the length of gene x, and n is the total number of predicted genes in each sample via CARD. The sum of the GCPM values for all the predicted genes in each sample was one million, making it comparable across the samples. To identify the association between the host bacteria and the ARGs, the NR database (accessed 01'25) was employed to map consensus of each ARG read sequence using the blastx (v2.16.0) and those with e-value less than $1ex10^{(-5)}$ were considered for potential associations. The Procrustes analysis was done to assess the concordance between microbial (Relative abundance) and gene expression (GCPM) datasets by superimposing their ordination plots using the 'protest()' function in R with 9,999 permutations. Each pair of points (microbe and gene) is connected by a line, with shorter lines indicating stronger agreement between the two datasets. The overall fit is quantified by the Procrustes sum of squares ($M^2$) and associated p-value, reflecting the strength and significance of the correlation. Further, the top ARGs host co-occurrence network was created based on the Spearman correlation in.gml format using the "networkx" library in Python and visualized and designed in Gephi (v0.10.1).

## Statistical analysis

The distribution of bacterial species between the COVID-19 and dengue patients was assessed using the Wilcoxon test using the R function "wilcox_test" of the "rstatix" package (v0.7.2) and beta diversity significance was calculated through PERMANOVA using "adonis2" function with permutations 999 of "vegan" package. Welch's t-test was used to calculate the significant differences among common species. Spearman correlations between ARGs and host bacteria were calculated using the "Spearmanr" function of the scipy package in Python. The distribution of common ARGs between COVID-19 and dengue was analyzed using the Wilcoxon test (R function "wilcox_test").

 

## Supporting information

**S1 File. Sensitivity analysis within the dengue cohort to assess and address potential sample type bias.**
(DOCX)

**S2 File. Alpha diversity (Chao1, Observed and Simpson indices) of ARGs and TAMs in COVID-19 and dengue infection.**
(DOCX)

**S3 File. Comparative analysis for concordance of ARGs and microbial abundance between two independent dengue cohorts for validation of study findings.**
(DOCX)

**S4 File. Genome coverage of dengue virus and SARS-CoV-2 across patient samples.**
(DOCX)

**S5 File. Evaluation of bracken-derived TAMs relative abundance estimates using multiple calculation methods.**
(DOCX)

**S6 File. Cumulative contribution of TAMs with >0.1% relative abundance to the total abundance within the corresponding phyla.**
(DOCX)

**S7 File. Distribution of microbial ARG coverage and multi-/single-site ARGs within the 30–50% threshold range.**
(DOCX)

**S1 Table. Antibiotic data for COVID-19 patients (n = 145).**
(XLSX)

**S2 Table. Gene Coverage Per Million (GCPM) values of antimicrobial resistance genes (ARGs) identified via CARD in COVID-19 and dengue patients.**
(XLSX)

**S3 Table. ARGs and their host TAMs in COVID-19 and dengue infection.**
(XLSX)

**S4 Table. Relative abundance of transcriptionally active microbes (TAMs) identified by Kraken2/Bracken2 in COVID-19 and dengue patients.**
(XLSX)

**S5 Table. Abundant ARGs (30% coverage and 10% sample occurrence) across COVID-19 and dengue severity classes.**
(XLSX)

**S6 Table. Logistic regression analysis of antimicrobial resistance genes (ARGs) in relation to comorbidities across COVID-19 severity groups.**
(XLSX)

## Acknowledgments

The authors duly acknowledge all the COVID-19 and dengue patients who participated in the study. Authors acknowledge the help and support from Dr. Aradhita Baral towards facilitation as research manager and coordination with the funders. The support of Lab manager Dr. Bharti Kumari is duly acknowledged for ensuring all reagents are available for the

experiments. Authors acknowledge the support of Anil Kumar, Nisha Rawat and Abhilash Thakur towards sample transport and sample management. We also acknowledge Ranjeet Maurya for uploading the RNA-Seq data onto NCBI. AY acknowledges CSIR, while JS and G acknowledge UGC for the fellowship support.

## Author contributions

**Conceptualization:** Rajesh Pandey.

**Data curation:** Raiyan Ali, Pallawi Kumari.

**Formal analysis:** Aanchal Yadav, Raiyan Ali, Priti Devi, Pallawi Kumari.

**Funding acquisition:** Rajesh Pandey.

**Investigation:** Aanchal Yadav, Raiyan Ali, Priti Devi, Pallawi Kumari, Jyoti Soni, Garima, Uzma Shamim, Rajesh Pandey.

**Methodology:** Aanchal Yadav, Priti Devi, Uzma Shamim.

**Project administration:** Rajesh Pandey.

**Resources:** Bansidhar Tarai, Sandeep Budhiraja, Rajesh Pandey.

**Supervision:** Rajesh Pandey.

**Visualization:** Aanchal Yadav, Raiyan Ali, Priti Devi, Pallawi Kumari.

**Writing – original draft:** Aanchal Yadav, Raiyan Ali, Priti Devi, Pallawi Kumari, Jyoti Soni, Garima.

**Writing – review & editing:** Uzma Shamim, Rajesh Pandey.

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
