## [Decision Letter · Decision Letter 0]

26 May 2025

Non-canonical Metatranscriptomic analysis of COVID-19 and Dengue reveals an expanded microbial and AMR landscape in COVID-19 mortality patients

PLOS Pathogens

Dear Dr. Pandey,

Thank you for submitting your manuscript to PLOS Pathogens. After careful consideration, we feel that it has merit but does not fully meet PLOS Pathogens's publication criteria as it currently stands. Therefore, we invite you to submit a revised version of the manuscript that addresses the points raised during the review process.

Please submit your revised manuscript within 60 days Jul 25 2025 11:59PM. If you will need more time than this to complete your revisions, please reply to this message or contact the journal office at plospathogens@plos.org. Please include the following items when submitting your revised manuscript:

We look forward to receiving your revised manuscript.

Kind regards,

Kevin Maringer, PhD

Academic Editor

PLOS Pathogens

Ashley St. John

Section Editor

PLOS Pathogens

Editor-in-Chief

PLOS Pathogens

orcid.org/0000-0003-2946-9497

Editor-in-Chief

PLOS Pathogens

orcid.org/0000-0002-7699-2064

**Additional Editor Comments:**

The reviewers noted the attention to an important research question and the potential significance of the data to the field, with potential broad interest across the virology and AMR communities. However, all reviewers also raised significant methodological concerns, and these will need to be addressed before the manuscript can be considered for publication in PLOS Pathogens. In particular, it will be crucial to include the additional methodological details requested, and to address concerns around the lack of controls, and quality control. All reviewers also raised important points about the metadata associated with the samples that were analysed, with significant attention required to control for the ways in which confounding factors such as antibiotic use and co-morbidities (amongst others) may affect the interpretation of the data.

**Journal Requirements:**

At this stage, the following Authors/Authors require contributions: Aanchal Yadav, Raiyan Ali, Priti Devi, Pallawi Kumari, Jyoti Soni, Garima Garima, Bansidhar Tarai, Sandeep Budhiraja, Uzma Shamim, and Rajesh Pandey. Please ensure that the full contributions of each author are acknowledged in the "Add/Edit/Remove Authors" section of our submission form.

https://journals.plos.org/plospathogens/s/submission-guidelines#loc-parts-of-a-submission

- ® on page: 34.

Potential Copyright Issues:

- Figures 1 and 3. Please confirm whether you drew the images / clip-art within the figure panels by hand. If you did not draw the images, please provide (a) a link to the source of the images or icons and their license / terms of use; or (b) written permission from the copyright holder to publish the images or icons under our CC BY 4.0 license. Alternatively, you may replace the images with open source alternatives. See these open source resources you may use to replace images / clip-art:

6) Please amend your detailed Financial Disclosure statement. This is published with the article. It must therefore be completed in full sentences and contain the exact wording you wish to be published. Please ensure that the funders and grant numbers match between the Financial Disclosure field and the Funding Information tab in your submission form. Note that the funders must be provided in the same order in both places as well. State what role the funders took in the study. If the funders had no role in your study, please state: "The funders had no role in study design, data collection and analysis, decision to publish, or preparation of the manuscript.".

**Reviewers' Comments:**

Reviewer's Responses to Questions

**Part I - Summary**

Reviewer #1: This is a well written manuscript that tackles and important and, mostly, unanswered question in AMR- how does the viral species (ssRNA based in this case) interact with the bacterial resistome. The study explored the burden of resistance between SARS-CoV-2 and Dengue patients and the microbiome, focusing on the transcriptionally active microbes and antimicrobial resistance genes.

The study uses large patient cohorts (COVID19- 251 patients, and Dengue 112 patients) which strengthens some of the statistical power and clinical relevance of the findings. However, there are some key limitations within the study that indicate some overstated claims without substantial evidence to support these with the data. there are some methodological limitations and lack of clarity with use of controls and possible biases in sampling.

However, the data sets are of interest to the AMR community and with major revisions the content would improve.

In terms of presentations, there is a lack of use of citations throughout to support methodologies and claims made. Adding suitable citations would scientifically improve the robustness of the manuscript. There are changes in font throughout the text, and use of abbreviations which have not been defined throughout. Addressing these would enhance readability.

Reviewer #2: I was pleased to have the opportunity to review this manuscript. This manuscript uses a metatranscriptomic approach to explore the role of the resistome and microbiome in viral infections, focusing on COVID-19 and dengue patients. It was great to see additional attempts at metatranscriptomic characterisation, given limited work done in this area with regard to the microbiome.

The use of metatranscriptomics to infer active microbial contributions adds novelty, and the dual-disease comparison is commendable. However, the manuscript requires clarification of methodological definitions, more rigorous justification of analytical choices (e.g., thresholds, tool appropriateness), and moderation of some interpretations pending supporting metadata. I provide specific recommendations below, and look forward to seeing the updated manuscript.

Reviewer #3: Study explores microbiome/resistome links to dengue and COVID-19 (with a range of clinical outcomes), an important and overlooked area, using “non-canonical meta-transcriptomics” (looking at transcriptionally active microbes?). A non-overlapping cohort of COVID/dengue was used, and different samples assayed (NP vs blood), from which RNAseq was carried out and analysed for ARGs by class etc (and later to bacterial species), and linked to clinical outcomes/compared between infections. This is important descriptive data, that highlights some interesting findings with potentially high relevance to clinical management and our understanding of the pathogenesis of viral diseases and their biology. From this one study, a lot of data are generated and many analyses are presented. However, serious questions remain about the study.

**Part II – Major Issues: Key Experiments Required for Acceptance**

Reviewer #1: The authors refer to a "non-canonical metatranscriptomics" approach to describe the methods used; however this term is undefined and not supported by a clear method when compared to standard metagenomic workflows and pipelines. Explaining this methodology, and its differences to standard metagenomics, would be helpful. There is no information about the specific bioinformatics software tools, benchmarking or validation of this approach; therefor it is not easy to assess the reproducibility and validity of this approach.

There is a lack of detail in the manuscript with regards to variables that are known to already influence the microbiome, for example associated with diet, antibiotic use and clinical information.

Detail on the antibiotic exposure, hospitalization durations, and ICU admission and comorbidities would enhance the findings of the study. At present, without this extra information, the direct linking of ARGs with viral disease does remain speculative. To improve, these factors should be accounted for.

There is evidence of possible sample bias in the study, where direct comparisons between clinical specimens derived from Covid19 patients (nasopharyngeal swabs) and dengue (blood samples) have been made; both have different microbiomes. It is currently unclear how sample origin affects resistome and microbiome composition, and thus this should be acknowledged in the text. Perhaps by using a sensitivity analysis where data is compared to type group only i.e. comparing results with dengue or covid19 cohorts respectively; or by testing for shared ARGs and taxa between both sample types.

At present much is inferred and claimed by the authors; but the above analyses would support more robust data interpretation.

Reviewer #2: The manuscript makes strong claims about β-lactam antibiotic use driving ARG prevalence, particularly stating: "its consistently high abundance in both infections suggests strong selective pressure from widespread β-lactam antibiotic use..." This inference is unsupported without patient-level antibiotic treatment data. If such metadata exist, they should be incorporated into the analysis. If not, the language must be moderated to avoid overinterpretation, focusing instead on ARGs as a potential reservoir.

The low threshold used for filtering ARGs (30% gene coverage and presence in at least 10% of samples) may reduce the reliability of the findings. The authors should provide a breakdown of how many ARGs surpass higher, more confident thresholds (e.g., ≥70% coverage), and discuss how the current threshold affects specificity.

The authors should consider mapping their sequencing reads to the Dengue virus and SARS-CoV-2 reference genomes to verify the presence of the primary pathogens. This serves as an important quality control measure to confirm that sequencing successfully captured host-pathogen dynamics and supports conclusions based on microbial transcriptional activity.

The use of Bracken for metatranscriptomic analysis should be justified more robustly or replaced. Bracken's assumptions of proportionality between read counts and species abundance are more appropriate for DNA-based metagenomics. Its use here may introduce systematic biases.

There is inconsistency and ambiguity in the use of the terms non-canonical metatranscriptomics and holo-transcriptomics. The manuscript must clearly define both terms and clarify whether they are interchangeable or distinct, and ensure consistent use throughout.

Reviewer #3: • The biggest one is that this is a cross-sectional study on two non-overlapping cohorts. Ideally, results would be validated in an independent cohort at least.

• There are issues with the description of their study, namely antibiotic usage patterns between the different diseases and severities. Do the authors have this? Similarly, were any patients co-infected with dengue and COVID?

• There is limited information on controls. The authors need to show data on negative controls for extraction and kits. Similarly, no healthy controls are assayed from relevant tissues.

**Part III – Minor Issues: Editorial and Data Presentation Modifications**

Reviewer #1: There are minor errors in regards to grammar and fonts throughout the text. More citations could be used throughout to support claims and methods used. Abbreviations are stated without clarification which leads to poor readability in eth text and in tables.

Reviewer #2: The statement in the Introduction: "The overuse of antibiotics, both in clinical settings and beyond, accelerates antimicrobial resistance (AMR), a crisis projected to cause 10 million deaths annually by 2050.” requires citation.

Genus and species names (e.g., Escherichia, Klebsiella) are not italicized consistently across the manuscript. This should be corrected in all sections, including the Conclusions.

The sentence "Building on our previous research on TAMs…” is vague. Please specify the findings from prior work and cite appropriately to provide context.

Inconsistent terminology for TAMs is used: "transcriptionally active microbial hosts” in the abstract and "metabolically or transcriptionally active ones” in the Introduction. These should be harmonized.

Some methodological content appears in the Results section. Please ensure all analytical steps and parameter descriptions are placed in the Methods and not repeated in Results unless essential for interpretation.

Figure 2D is difficult to interpret. The heatmaps appear split (e.g., 2D-i) with boxed genes and no explanation for separation or grouping. Labeling must be clarified, and the figure legend should explain how heatmaps are structured and what the boxes denote.

The statement: "847 and 40 ARGs exclusive to COVID-19 and dengue..." lacks sufficient detail on detection confidence. This is partly addressed in Part II but should also be reflected in the figure or text to guide reader interpretation.

Formatting of software tools is inconsistent. FastQC is mentioned without a version, while Trimmomatic includes a version. Please align formatting for clarity.

The Methods section mentions the use of QIAGEN kits with "specific adjustments, including..." but does not list all modifications. Either enumerate all adjustments or revise the sentence to avoid implying there are unlisted changes.

The thresholds used to define TAMs (e.g., >0.1% for species) are relatively low. Though this point is related to method justification, a brief rationale should be added or referenced to support their choice.

The claim: "this consistency reinforces the idea that while ARGs play a crucial role in COVID-19 severity, they have minimal impact on dengue progression” could be misleading. It is also possible that COVID-19 patients had more antibiotic exposure. This point is acknowledged later in the manuscript, but it should be flagged here as an interpretative limitation.

The final paragraph of the Conclusions could benefit from a brief note on limitations (e.g., lack of metadata, challenges of functional inference from RNA data) and a comment on the applicability of this method to public health surveillance (cost, scalability).

Reviewer #3: (No Response)

PLOS authors have the option to publish the peer review history of their article (what does this mean? ). If published, this will include your full peer review and any attached files.

**Do you want your identity to be public for this peer review?** For information about this choice, including consent withdrawal, please see our Privacy Policy .

Reviewer #1: No

Reviewer #2: **Yes: ** Edward Cunningham-Oakes

Reviewer #3: No

**Figure resubmission:**

**Reproducibility:**



---

## [Decision Letter · Decision Letter 1]

9 Nov 2025

Dear Dr. Pandey,

We are pleased to inform you that your manuscript 'Non-canonical Metatranscriptomic analysis of COVID-19 and Dengue reveals an expanded microbial and AMR landscape in COVID-19 mortality patients' has been provisionally accepted for publication in PLOS Pathogens.

Best regards,

Kevin Maringer, PhD

Academic Editor

PLOS Pathogens

Ashley St. John

Section Editor

PLOS Pathogens

Sumita Bhaduri-McIntosh

Editor-in-Chief

PLOS Pathogens

orcid.org/0000-0003-2946-9497

Michael Malim

Editor-in-Chief

PLOS Pathogens

orcid.org/0000-0002-7699-2064

The reviewers recognised the additional analysis and rewrites of the manuscript included in this revision, and are in agreement that the study is much improved. You will find one of the reviewer's responses missing, but having read the response to the reviewers' comments in detail myself, I am in agreement with their assessment on the improved quality of the manuscript.

Reviewer Comments (if any, and for reference):

Reviewer's Responses to Questions

**Part I - Summary**

Reviewer #2: The authors have addressed all concerns raised in the initial review with care and precision. The revised manuscript is well-structured, methodologically sound, and provides comprehensive benchmarking that clarifies and validates their approach.

Reviewer #3: The authors appear to have addressed all my concerns. Thank you.

**Part II – Major Issues: Key Experiments Required for Acceptance**

Reviewer #2: None. The authors have convincingly addressed prior concerns, and I find no major flaws remaining in the manuscript.

Reviewer #3: The authors appear to have addressed all my concerns. Thank you.

**Part III – Minor Issues: Editorial and Data Presentation Modifications**

Reviewer #2: None to note.

Reviewer #3: The authors appear to have addressed all my concerns. Thank you.

PLOS authors have the option to publish the peer review history of their article (what does this mean? ). If published, this will include your full peer review and any attached files.

**Do you want your identity to be public for this peer review?** For information about this choice, including consent withdrawal, please see our Privacy Policy .

Reviewer #2: **Yes: ** Edward Cunningham-Oakes

Reviewer #3: No